# Accelerating Vision Transformers with Adaptive Patch Sizes

**Rohan Choudhury**[1*]    **JungEun Kim**[2,3*]    **Jinhyung Park**[1]
**Eunho Yang**[2]    **László A. Jeni**[1†]    **Kris M. Kitani**[1†]
[1]Carnegie Mellon University    [2]KAIST    [3]General Robotics

## Abstract

Vision Transformers (ViTs) partition input images into uniformly sized patches regardless of their content, resulting in long input sequence lengths for high-resolution images. We present Adaptive Patch Transformers (APT), which addresses this by using multiple different patch sizes within the same image. APT reduces the total number of input tokens by allocating larger patch sizes in more homogeneous areas and smaller patches in more complex ones. APT achieves a drastic speedup in ViT inference and training, increasing throughput by 40% on ViT-L and 50% on ViT-H while maintaining downstream performance. It can be applied to a previously fine-tuned ViT and converges in as little as 1 epoch. It also significantly reduces training and inference time without loss of performance in high-resolution dense visual tasks, achieving up to 30% faster training and inference in visual QA, object detection, and semantic segmentation. Our project page is available at this link.

## 1 Introduction

Vision Transformers (ViTs) (Dosovitskiy et al., 2020) have become the dominant paradigm for visual recognition, but their scalability is limited by the quadratic cost of self-attention with respect to sequence length. Since inputs are divided into fixed-size patches, image resolution directly determines sequence length: higher resolution images yield disproportionately long token sequences despite much higher redundancy.

Many prior works have proposed solutions to this issue, typically by merging a fixed proportion of similar tokens (Bolya et al., 2022) or pruning uninformative ones with auxiliary predictors (Rao et al., 2021; Yin et al., 2022). While these reduce theoretical FLOPs, they face two drawbacks. Firstly, a fixed reduction ratio is mismatched to image complexity: merging only half the tokens in a pure white image is insufficient, while merging half the tokens in a busy cityscape is harmful. Secondly, pruning during the forward pass introduces padding and irregular shapes, often negating speedups in practice (Dehghani et al., 2021). In contrast to vision transformers, language models rely on adaptive tokenizers such as Byte-Pair Encoding (Sennrich et al., 2016) and SentencePiece Kudo & Richardson (2018), which flexibly assign tokens of varying lengths depending on subword frequency. This reduces input sequence size while improving performance, suggesting that variable-granularity tokenization can be more efficient than fixed-size splits.

Our key insight is that a similar idea can be applied to vision transformers. As illustrated in Figure 1, ViTs use the same amount of computation on a uniform green background as on the complex patches on the head of the bird, despite the significant difference in visual complexity. We introduce the Adaptive Patch Transformer (APT), which addresses this mismatch by varying patch sizes *within a single image*. Regions that are smooth and redundant can be represented with large patches, while regions rich in detail are allocated smaller patches. This content-aware patchification preserves important information where it matters while reducing redundancy elsewhere. To do this, APT computes entropy at multiple scales and assigns larger patch sizes to regions with the lowest entropy,

---

*Equal contribution
†Equal advising.

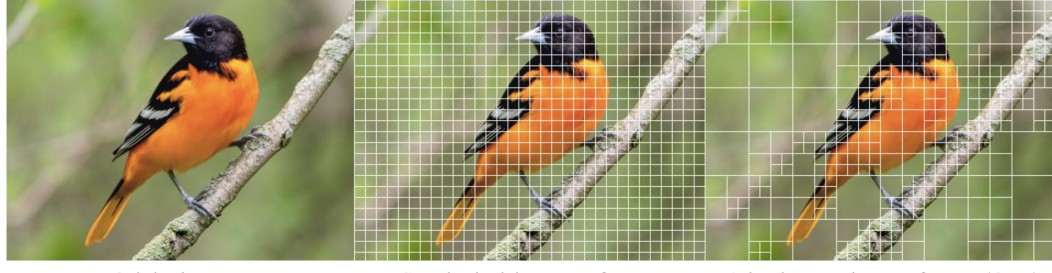

| Original Image | Standard Vision Transformer | Adaptive Patch Transformer (Ours) |

Figure 1: **Adaptive Patch Sizing.** We present APT, Adaptive Patch Transformers, which significantly accelerate vision transformer training and inference by patchifying images based on their content. Complex regions receive more, smaller tokens, while simpler, homogeneous regions receive fewer.

resulting in significantly fewer input tokens. We then down-sample the larger patches and combine their patch embeddings with the information from the original large patch using a zero-initialized MLP, allowing APT to converge without harming the network.

APT speeds up ViT inference *and* training by almost 40%, with even larger boosts for higher resolution images and larger models. When initialized from a self-supervised or large-scale pretrained checkpoint, APT reaches the same performance as the original ViT after fine-tuning. If applied directly to an already fine-tuned ImageNet checkpoint, APT incurs only a small accuracy drop without additional training. With our zero-initialized MLP, this gap can be closed in as little as a single epoch of fine-tuning. We also find that unlike most prior token reduction works, APT can successfully accelerate vision transformers on a wide range of image understanding tasks, such as visual question answering, object detection, and semantic segmentation, while matching the baseline performance.

In summary, we (1) introduce the Adaptive Patch Transformer (APT), which accelerates Vision Transformers by up to 40% through content-aware patch sizes, with larger gains at higher resolutions and model scales; (2) show that APT preserves the accuracy of standard pretrained models across resolutions and scales; and (3) demonstrate that APT extends beyond ImageNet, performing well on dense prediction and vision-language tasks.

## 2 RELATED WORK

**Vision Transformers and Patchification.** Vision Transformers (ViTs) (Dosovitskiy et al., 2020) are currently the *de facto* standard architecture for computer vision backbones (Xu et al., 2022; Kirillov et al., 2023; Peebles & Xie, 2022). In contrast to language models, which typically use subword tokenizers (Sennrich et al., 2016; Kudo & Richardson, 2018) with varying numbers of bits per token, ViTs *patchify* images into equally sized patches, each becoming a token. This can result in an enormous number of tokens, especially at high resolution. Transformer-based generative models (Peebles & Xie, 2022; Esser et al., 2020) use visual tokenizers, typically using a variational auto-encoder (Kingma et al., 2013; Van Den Oord et al., 2017), to project images into a compressed latent space, reducing the input size significantly. Some recent works explore adaptive visual tokenizers (Yan et al., 2024; Duggal et al., 2024), which dynamically allocate more tokens to more complex visual inputs, but do not meaningfully speed up training or generation. As a result, image understanding tasks are typically limited to lower resolution.

**Reducing ViT Tokens.** Accelerating ViTs by removing tokens is a rich area of research. Methods such as pruning (Yu & Xiang, 2023; Yang et al., 2023; Zheng et al., 2022), compressed representations (Wu et al., 2018; Park & Johnson, 2023), or quantization (Liu et al., 2021b; Li et al., 2022c; Moon et al., 2024) remove redundancies or compactly encode parameters, reducing inference time and memory usage. Alternative attention mechanisms, such as linearized (Katharopoulos et al., 2020; Lu et al., 2021) or local window attention (Liu et al., 2021a; Wei et al., 2023; Chen et al., 2023b), improve efficiency by limiting token interactions. More related to our work are methods that exploit the inherent redundancy of images (Meng et al., 2022; Yin et al., 2022; Kong et al., 2022; Rao et al., 2021) and videos (Choudhury et al., 2025; Ding et al., 2023; Wu et al., 2023) by

pruning uninformative tokens. While these works are content-aware, most require learning which tokens are unhelpful, negating any training speedup and preventing inference on batch sizes greater than 1. Several methods instead merge tokens based on similarity (Bolya et al., 2022; Bolya & Hoffman, 2023; Liang et al., 2022b; Shang et al., 2024; Cao et al., 2023; Liang et al., 2022a; Tran et al., 2024; Kallini et al., 2024; Lee & Hong, 2024; Wu et al., 2023), which accelerates training. However, merging methods typically combine a constant number of tokens for each input, which can be suboptimal for inputs with varying complexities. APT strikes a balance between these two lines of work by providing significant acceleration to training and inference while maintaining content-awareness.

**Adaptive Patch Sizing for Efficient ViTs.** Our work is not the first to propose using multiple patch sizes for faster ViTs. Early attempts in this direction (Chen et al., 2021; Beyer et al., 2023; Wang et al., 2024; 2021; Zhou & Zhu, 2023; Hu et al., 2024) train models that are capable of using different patch sizes, but still require a single patch size for each image. Closer to APT are works that allow for varying patch sizes within a single image (An et al., 2024; Ronen et al., 2023; Chen et al., 2023a; Bai et al., 2024). However, CF-ViT (Chen et al., 2023a) and Quadformer (Ronen et al., 2023) rely on a fixed number of patches, neglecting the variability of semantic information across images, which can lead to suboptimal performance. MG-VIT (Zhang et al., 2023b) also supports two patch scales, but relies on attention scores to decide patch sizes, preventing use of efficient attention kernels, and also requires training from scratch, which is significantly more expensive than APT.

Closest to our work is MS-ViT (Havtorn et al., 2023), which, like DynamicViT (Rao et al., 2021) learns a gating network to determine patch sizes and defines separate patch embedding networks for each size. However, it requires significant fine-tuning on pre-trained networks and does not speed up training. APT resolves this issue while demonstrating dramatically larger speedups at higher resolutions and on larger models.

## 3 METHOD

Our goal is to achieve a significant wall-clock speedup during *both* training and inference by using different-sized patches in different regions of the image. We first describe how we allocate different patch sizes within an image (Section 3.1) and then how we process different-sized regions into the same embedding space (Section 3.2). We then explain how we efficiently handle different input sizes and how we can adapt APT to work on dense visual prediction tasks like object detection.

### 3.1 DECIDING PATCH SIZES

Consider a vision transformer that takes an $H \times W \times C$ image as input. The standard ViT partitions the image into a set of $p \times p$ patches. A linear layer $\mathcal{E}$ is applied to each patch to convert it into a token, of size $d_{embed}$, resulting in a sequence of $N = (HW/p^2)$ tokens.

In contrast, our goal is to decide patch size based on the image content, instead of using a constant number of patches. Concretely, we define a fixed number of patch scales $S$, where the set of patches consists of $\mathcal{P} = P_1 \cup P_2 \cup \ldots P_S$, with each patch in $P_i$ having size $2^i p \times 2^i p$. For example, if $S = 3, p = 16$, we are trying to find a smaller set of $16 \times 16, 32 \times 32$ and $64 \times 64$ patches while maximizing 'information' conveyed. For simplicity, we also impose the constraint that all patches follow a quadtree-like structure following a regular grid.

We use *entropy* $H$ as a measure of a patch's compressibility, given by:

$$H(P) = -\sum_{i=0}^{L-1} p_i \log_2 p_i, \tag{1}$$

where $p_i$ is the probability of pixel intensity $i$. Since patches contain discrete pixel values, we approximate this by binning pixel intensities and computing entropy from the resulting distribution. Entropy quantifies the unpredictability and thus information content of a patch, making it a useful predictor of compressibility—lower entropy indicates higher redundancy. A large patch with low entropy should therefore be efficiently representable by a $d_{embed}$-dimensional vector. We discuss alternative measures further in the Appendix.

We obtain the patchification of the image hierarchically, as illustrated in Figure 3. We first divide the image into patches at the coarsest scale $2^S p \times 2^S p$ and compute their entropies. We then retain

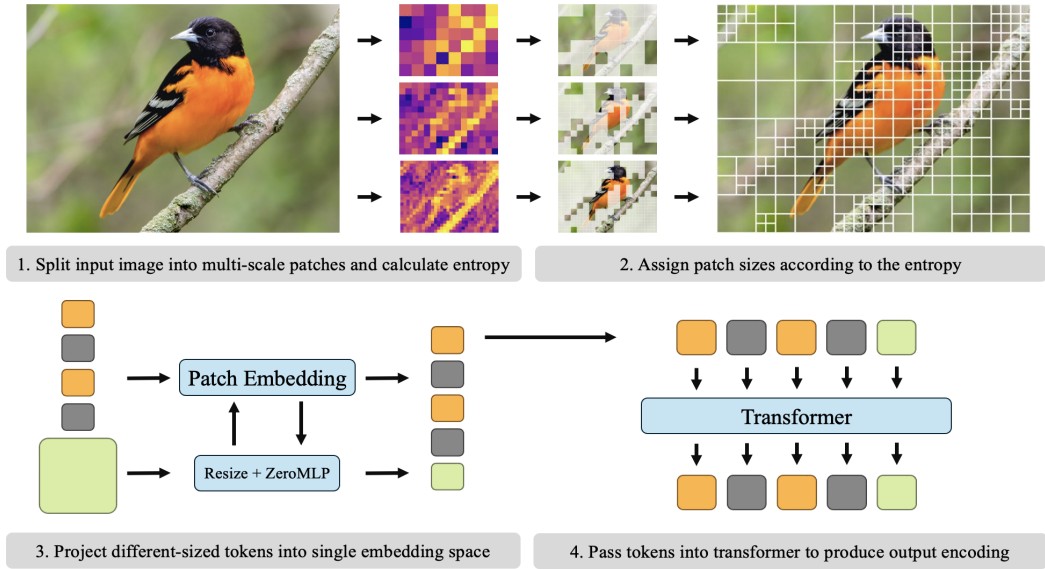

Figure 2: **APT overview.** APT works by measuring the entropy at multiple scales and assigning large patch sizes to low entropy patches. All patches are projected to the same size token embedding, and the reduced size input sequence is passed to the transformer.

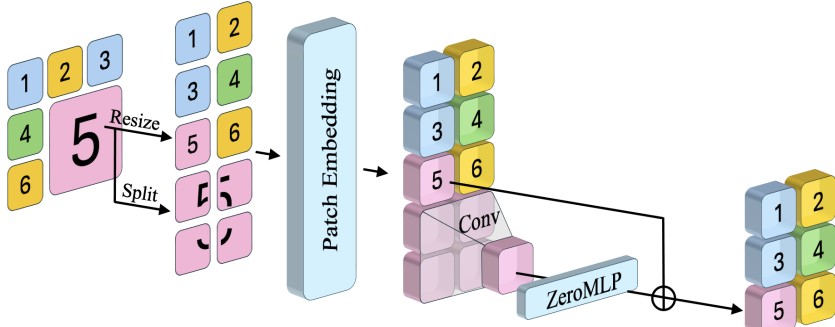

Figure 3: **Embedding Different Patch Sizes.** The smallest size patches are projected with the patch embedding. Larger patches are both split into their sub-patches and resized; the sub-patches are embedded, aggregated with a convolution layer. These are combined with the resized embedding with a zero-initialized MLP (Zhang et al., 2023a).

all such patches with entropy below a fixed threshold $\tau_i$, which is a tunable hyperparameter for each level. We repeat this process until we reach the smallest possible patch size $p \times p$, to which we assign all remaining patches.

## 3.2 PATCH AGGREGATION

After dividing the image into different patch sizes, we need to convert these patches into embeddings with dimension $d_{\text{embed}}$; in standard ViTs, this is done with a single linear layer $\mathcal{E}$. Prior work on vision transformers with varying patch sizes either resize every patch to $p \times p$ (Ronen et al., 2023), resize $\mathcal{E}$ for each size (Beyer et al., 2023), or train $S$ separate patch embedding layers $\mathcal{E}_i$ for each possible patch size (Havtorn et al., 2023), adding overhead. Resizing allows reasonable performance with no training but can be improved upon—it uses strictly less information than if we applied $\mathcal{E}$ to the higher-resolution sub-patches.

We combine these strategies, as shown in Figure 3. We resize patches to a uniform size $p \times p$ but retain copies of larger original patches. For a given patch $P_i$ of size $2^i p \times 2^i p$, we define its constituent $p \times p$ sub-patches as the set $\{P_j\}$. Each sub-patch $P_j$ is embedded using the standard

embedding layer $\mathcal{E}$. The final embedding for patch $P_i$ is then computed as:

$$\mathcal{E}(P_i) = \text{ZeroMLP}\left(\text{Conv2d}^{(i)}(\{\mathcal{E}(P_j) \mid P_j \subset P_i\})\right) + \mathcal{E}(\text{Resize}_p(P_i)), \quad\quad (2)$$

where $\text{Conv2d}^{(i)}$ indicates applying a convolutional downsampling layer $i$ times, aggregating embeddings from sub-patches back to size $p \times p$. The ZeroMLP, a single linear layer initialized with zero weights inspired by ControlNet (Zhang et al., 2023a), allows the model to gradually incorporate high-resolution details without initially degrading performance, facilitating faster convergence during fine-tuning. In particular, this enables APT to be applied to any pre-trained ViT and matches the performance of the initial model with a single epoch of accelerated fine-tuning.

## 3.3 DYNAMIC INPUT SIZES

Since APT is content-aware, the number of tokens for each image can vary widely. However, in contrast to token pruning works (Rao et al., 2021; Liang et al., 2022b), we do not reduce the size of the input at each layer, but *before* running the model. While most vision works use a fixed resolution, our setting is closer to that of language modeling, and in vision to RLT (Choudhury et al., 2025) and NaViT (Dehghani et al., 2023), where the number of tokens varies, but is predictably dictated by the input data. We follow these methods and employ sequence packing. For a batch of input images with sequence lengths $\{N_1, N_2, \ldots N_B\}$, we concatenate the tokens into a single sequence with length $\sum_{i=1}^{B} N_i$ and construct a block-diagonal mask that ensures tokens only attend to tokens from the same example. This is natively implemented in commonly available attention backends such as FlashAttention (Dao et al., 2022; Dao, 2024) or xFormers (Lefaudeux et al., 2022), and adds no overhead to the network itself as the mask does not change. After running the network, we split the resulting sequence into its constituent subsequences and either extract the class token or compute a pooled representation for each subsequence.

**Positional Encodings.** To handle the positional encodings for the new variable-length sequences, we use positional encoding interpolation, introduced in NaViT (Dehghani et al., 2023). Each smallest-size patch grid of size $\frac{H}{p} \times \frac{W}{p}$ is assigned an initial positional encoding, whether learned, sinusoidal or RoPE, where $p$ is the base patch size. For larger patch sizes, we obtain the corresponding positional encodings through interpolation: patches of size $sp$ (where $s > 1$) use a $\frac{H}{sp} \times \frac{W}{sp}$ grid, whose positional encodings are computed by sampling from the original $\frac{H}{p} \times \frac{W}{p}$ encoding map. For example, patches of size $2p$ use encodings interpolated to an $\frac{H}{2p} \times \frac{W}{2p}$ grid, patches of size $4p$ use $\frac{H}{4p} \times \frac{W}{4p}$, and so on. This approach naturally generalizes positional information across different scales while maintaining spatial consistency.

**Adaptation to Downstream Tasks.** Standard methods for dense visual tasks like object detection or semantic segmentation often rely on a feature map that has the same aspect ratio as the image. This is required for methods that rely on transposed convolutions to upsample an input feature map for per-pixel predictions (Li et al., 2022a). In contrast, APT produces a different number of tokens per image, which cannot be simply reshaped into a rectangular feature map. To handle this, we rely on the assumption that the tokens representing larger patches encode simpler features and simply repeat them $2^{2i}$ times, as in (Havtorn et al., 2023; Bolya & Hoffman, 2023). This yields a fully differentiable feature map that can be upsampled by transposed convolutions and seamlessly applied to downstream tasks. Furthermore, tasks requiring high-resolution dense prediction, such as object detection, often rely on *window attention* (Liu et al., 2021a; Yuan et al., 2021; Fang et al., 2024), where the image is subdivided into multiple window regions to localize attention and increase efficiency. APT can still be applied even with window-attention. To do this, we divide the image into windows that are multiples of the largest patch-size, and apply our patch assignment strategy as before. Now, each window contains variable numbers of tokens rather than a constant number, and attention is applied within each window. As before, this can be straightforwardly implemented using sequence packing and attention masks with light overhead.

| Model | Res/Patch | Acc ↑ | Img/s ↑ | GFLOPS ↓ | WC Time ↓ | Speedup ↑ |
|---|---|---|---|---|---|---|
| ViT-B$^{MAE}$ | 384/16 | 84.2 | 1151 | 49.4 | 11.6h | - |
| Random | 384/16 | 83.4 | 1401 | 21.5 | 8.8h | +32% |
| Resizing | 384/16 | 83.9 | 1390 | 21.5 | 9.0h | +29% |
| **APT (Ours)** | 384/16 | 84.2 | 1390 | 21.9 | 9.0h | +29% |
| ViT-L$^{MAE}$ | 336/14 | 86.1 | 395 | 174.7 | 15.9h | - |
| Random | 336/14 | 85.5 | 550 | 76.2 | 9.6h | +66% |
| Resizing | 336/14 | 85.9 | 527 | 76.2 | 9.9h | +61% |
| **APT (Ours)** | 336/14 | 86.1 | 527 | 76.8 | 9.9h | +61% |
| ViT-L$^{MAE}$ | 448/14 | 86.4 | 190 | 645 | 31.4h | - |
| Random | 448/14 | 85.8 | 314 | 267 | 16.2h | +94% |
| Resizing | 448/14 | 86.0 | 302 | 267 | 16.9h | +86% |
| **APT (Ours)** | 448/14 | 86.3 | 302 | 268 | 16.9h | +86% |

Table 1: **Full Fine-Tuning on ImageNet.** APT significantly reduces the wall-clock time to fine-tune a pre-trained backbone on ImageNet with no degradation in accuracy. We use the MAE (He et al., 2021) training recipe for all cases. Note that ViT-B is trained for 2× more epochs than ViT-L.

| Model | Res/Patch | Acc ↑ | GFLOPS ↓ | Img/s ↑ | Speedup ↑ | Res/Patch | Acc ↑ | GFLOPS ↓ | Img/s ↑ | Speedup ↑ |
|---|---|---|---|---|---|---|---|---|---|---|
| ViT-B | 224/16 | 85.1 | 16.9 | 3310 | - | 384/16 | 86.1 | 49.4 | 1151 | - |
| Random | 224/16 | 83.7 | 12.5 | 3751 | +13% | 384/16 | 85.0 | 21.5 | 1401 | +22% |
| Resizing | 224/16 | 84.6 | 12.5 | 3540 | +7% | 384/16 | 85.7 | 21.5 | 1390 | +21% |
| **APT-B (Ours)** | 224/16 | 85.1 | 12.7 | 3540 | +7% | 384/16 | 86.1 | 21.9 | 1390 | +21% |
| ViT-L | 224/14 | 87.9 | 59.7 | 883 | - | 336/14 | 88.2 | 174.7 | 395 | - |
| Random | 224/14 | 86.9 | 44.3 | 1049 | +19% | 336/14 | 87.3 | 76.2 | 550 | +39% |
| Resizing | 224/14 | 87.4 | 44.3 | 993 | +12% | 336/14 | 87.9 | 76.2 | 527 | +33% |
| **APT-L (Ours)** | 224/14 | 87.8 | 44.5 | 993 | +12% | 336/14 | 88.1 | 76.8 | 527 | +33% |
| ViT-H | 224/14 | 88.3 | 162.0 | 441 | - | 336/14 | 88.5 | 363.7 | 175 | - |
| Random | 224/14 | 87.4 | 92.1 | 568 | +29% | 336/14 | 87.0 | 158.3 | 272 | +55% |
| Resizing | 224/14 | 88.0 | 92.1 | 542 | +23% | 336/14 | 88.0 | 158.3 | 263 | +50% |
| **APT-H (Ours)** | 224/14 | 88.3 | 92.3 | 542 | +23% | 336/14 | 88.4 | 158.9 | 263 | +50% |

Table 2: **1-epoch Fine-Tuning on ImageNet.** APT consistently achieves large speedups while matching or sometimes exceeding the original network's performance after fine-tuning for 1 more epoch. Compared to only random masking or only resizing, APT offers the best tradeoff between speed and accuracy.

## 4 EXPERIMENTS

### 4.1 BASELINES

We categorize token merging approaches into two groups: *input-level* and *layer-level*. Input-level merging reduces tokens directly from image patches before entering the model, which is the category our method belongs to. In contrast, layer-level merging performs reduction within the network during feature propagation. We adopt input-level merging as our main baseline for fairest comparison, but compare to layer-level methods as well.

**Input-level Merging Baselines.** We use three main baselines: *random-masking*, *resizing-only* (He et al., 2021; Li et al., 2023) and the original optimized *Vanilla* implementation from `timm`. *Resizing* refers to only resizing the larger patches to the base patch size; This is a stronger version of Quad-former (Ronen et al., 2023), which also resized the larger patches, but used a constant, nonadaptive number of patches per image; instead we use a threshold so that the sequence length is adaptive. *Random* is a stronger version of FLIP (Li et al., 2023); we compute the token reduction obtained from APT and set it as the random patch dropping rate.

**Layer-level Merging Baselines.** We benchmark against four baselines: EViT (Liang et al., 2022a), ToMe (Bolya et al., 2022), PPT (Wu et al., 2023) and DTEM (Lee & Hong, 2024). These baselines perform token merging across the ViT layers, removing a constant number of tokens regardless of image content. However, they share a key limitation: none of them are natively compatible with FlashAttention, which makes them even slower than the Vanilla ViT equipped with FlashAttention. To provide a fairer and stronger comparison, we re-implement 'advanced' versions of these baselines with FlashAttention, with the exception of PPT, which relies on attention scores and is thus

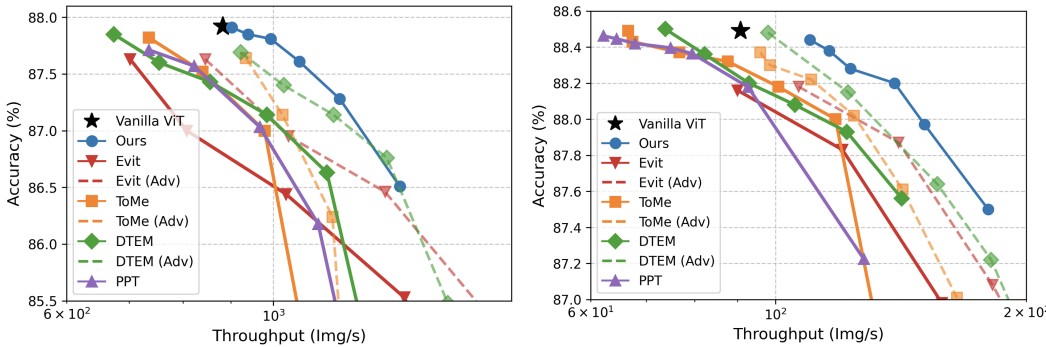

(a) Trade-off comparison on ViT-L/14, $224 \times 224$.  (b) Trade-off comparison on ViT-H/14, $336 \times 336$.

Figure 4: **Accuracy vs. Throughput under different compute budgets.** Comparison between APT and layer-level merging methods on ViT-L and ViT-H. For a fairer evaluation, we also include their re-implemented Advanced (Adv) versions with FlashAttention, shown with a dashed line. APT consistently outperforms the baselines in both throughput and accuracy across all compute budgets.

incompatible. For methods that use weighted attention, we disable it to enable FlashAttention. We provide more details in Appendix A.

## 4.2 IMAGE CLASSIFICATION

**Full Fine-Tuning.** We provide the results of fine-tuning a vision transformer with APT on two different model scales and resolutions in Table 1. Here, 'full' fine-tuning refers to training from a pre-trained self-supervised ViT backbone, rather than an already fine-tuned classification network. In all experiments, we use the official MAE (He et al., 2021) pre-trained checkpoint, interpolated to match the target resolution. At lower resolution, APT provides a ∼10% speedup over the baseline with a ∼14% token reduction. However, the speedup increases dramatically for higher resolutions—on 336×336, the speedup doubles. APT also reduces training time more at larger model scales, likely due to the fact that the attention computation dominates much more of the training time per iteration. In all cases, APT matches the baseline while using the exact same training recipe—it can be considered an absolute improvement over standard patching.

**Short Fine-Tuning.** We next present results from training with APT for 1 epoch starting from a checkpoint already fine-tuned on ImageNet (Deng et al., 2009), rather than a pre-trained self-supervised checkpoint. Compared to other methods like DynamicVIT (Rao et al., 2021) or MS-VIT (Havtorn et al., 2023) which require 50 or more epochs of fine-tuning to learn a scoring function, thanks to our use of a zero-initialized layer, APT models make high-quality predictions from initialization. With no training, APT only resizes the larger patches to the base size, which is a stronger version of Quadformer (Ronen et al., 2023). However, we observe that just one epoch is sufficient to "heal" the degradation from the new patchification scheme and match the original performance of the model, as shown in Table 2.

We provide a comparison with representative baselines in Figure 4. We compare throughput and ImageNet accuracy to our short fine-tuning results on ViT-L/14 with a resolution of 224 and ViT-H/14 with a resolution of 336. APT consistently outperforms all baselines, including their original versions as well as our improved reproductions using FlashAttention. The results confirm that input-level merging is inherently more efficient and reliable than layer-level merging.

## 4.3 VQA AND DENSE VISUAL TASKS

Vision transformers are used for a wide range of tasks beyond image classification; we evaluate how APT affects downstream performance in vision-language understanding tasks as well as dense prediction. Building on our fine-tuning experiment for image classification, we start with a fully fine-tuned model, and fine-tune for 5% of the total iterations used in each model's fine-tuning scheme. For Visual QA, we fine-tune only the newly introduced ZeroConv and MLP modules, while for detection and segmentation, we fine-tune the entire model.

| Model | Img/s | Speedup | $VQA^{v_2}$ | GQA | $SQA^I$ | $VQA^T$ | POPE | MME | MMB | $MMB^C$ | MMV |
|---|---|---|---|---|---|---|---|---|---|---|---|
| LLaVA-1.5-7B | 3.70 | - | 78.5 | 61.0 | 67.8 | 58.2 | 86.9 | 1510.1 | 64.6 | 58.1 | 30.7 |
| Random | 4.58 | +24% | 76.9 | 60.9 | 67.2 | 54.1 | 86.1 | 1460.5 | 62.7 | 57.6 | 30.5 |
| Resizing | 4.51 | +22% | 77.5 | 61.1 | 66.8 | 56.5 | 86.6 | 1473.8 | 63.2 | 58.1 | 30.2 |
| **APT (Ours)** | 4.51 | +22% | 77.9 | 61.4 | 67.5 | 56.9 | 86.4 | 1474.0 | 63.8 | 58.2 | 30.8 |
| LLaVA-1.5-13B | 2.22 | - | 80.0 | 63.2 | 72.7 | 61.2 | 87.1 | 1530.6 | 68.5 | 63.4 | 35.4 |
| Random | 2.79 | +26% | 78.0 | 60.7 | 72.0 | 55.7 | 86.5 | 1484.0 | 64.7 | 60.8 | 32.0 |
| Resizing | 2.72 | +23% | 78.9 | 61.1 | 72.0 | 59.1 | 86.8 | 1496.9 | 65.8 | 62.5 | 33.9 |
| **APT (Ours)** | 2.72 | +23% | 79.4 | 63.0 | 72.4 | 59.5 | 87.2 | 1511.2 | 66.5 | 63.7 | 34.7 |

Table 3: **Transfer to VQA**. APT enables significant throughput increase while matching or exceeding performance to the baseline.

| Model | Res | Img/s | Speedup | mAP | AP50 |
|---|---|---|---|---|---|
| EVA-02-B | 1536 | 3.86 | - | 58.93 | 77.85 |
| Resizing | 1536 | 4.41 | +14% | 58.43 | 77.22 |
| **APT (Ours)** | 1536 | 4.41 | +14% | 58.79 | 77.65 |
| EVA-02-L | 1536 | 1.62 | - | 62.28 | 80.80 |
| Resizing | 1536 | 2.17 | +30% | 61.75 | 80.27 |
| **APT (Ours)** | 1536 | 2.17 | +30% | 62.07 | 80.64 |

| Model | Res | Img/s | Speedup | aAcc | mIoU |
|---|---|---|---|---|---|
| EVA-02-L | 512 | 4.40 | - | 86.67 | 59.77 |
| Resizing | 512 | 4.87 | +11% | 86.09 | 58.81 |
| **APT (Ours)** | 512 | 4.87 | +11% | 86.68 | 59.70 |
| EVA-02-L | 640 | 2.55 | - | 86.83 | 60.05 |
| Resizing | 640 | 2.83 | +11% | 86.06 | 58.83 |
| **APT (Ours)** | 640 | 2.83 | +11% | 86.82 | 60.01 |

Table 4: **Transfer to Object Detection.** APT can be scaled to high-resolution dense image tasks supporting window attention.

Table 5: **Transfer to Semantic Segmentation.** APT can handle pixel-level fine-grained tasks without compromising visual acuity.

**Visual QA.** We first apply APT to the vision backbone of LLaVA (Liu et al., 2023; 2024a). LLaVA is a vision language model (VLM) that combines a vision transformer backbone with a language backbone via a projection layer. In the original paper (Liu et al., 2023), the vision encoder was completely frozen, and only the projection layer was updated. APT matches the original model performance while reducing image tokens and increasing throughput by 23%. Note that APT provides no speedup to the language component, but by reducing the number of visual tokens, it accelerates both the vision backbone and cross attention layers. We find that APT exceeds the original performance of the LLaVA model on a range of vision-language benchmarks (Goyal et al., 2017; Hudson & Manning, 2019; Lu et al., 2022; Singh et al., 2019; Schwenk et al., 2022; Fu et al., 2024; Liu et al., 2024b; Yu et al., 2023).

**Object Detection.** One might expect APT to degrade performance for tasks that require pixel-level understanding, such as object detection. To investigate this, we trained an object detector using the EVA-02 (Fang et al., 2024) backbone with window attention, with a ViTDet (Li et al., 2022b) style detection head. We conduct experiments on the COCO (Lin et al., 2014) dataset at 1536 × 1536 resolution. APT is able to reduce an impressive 30% of input tokens, drastically speeding up training and inference, while matching the final performance on mAP and AP50. Furthermore, these results demonstrate that APT remains effective under window attention beyond naive full attention, broadening the scope of its application.

**Semantic Segmentation.** We conduct another experiment on semantic segmentation, which requires fine-grained understanding of object boundaries. We again use the protocol of EVA-02 (Fang et al., 2024), using it as a backbone with a UperNet (Xiao et al., 2018) segmentation model on top. When tested on ADE20K (Zhou et al., 2019; 2017), APT attains baseline performance while reducing 28~32% of the input tokens depending on image resolution, thereby substantially accelerating inference. APT's success at semantic segmentation is particularly encouraging, since it implies that it reduces compute while not sacrificing visual acuity at the pixel level.

## 4.4 ABLATIONS

We ablate components of APT to evaluate their effect on speed and accuracy.

**Measuring APT overhead.** Next, we measure the computational overhead introduced by APT. Rearranging the input patches and using masks does not have zero computational cost, and given that GPUs are highly optimized for constant input shapes, understanding the cost of adding APT is important. The results of this analysis are in Table 6, showing that with no sequence reduction, APT is

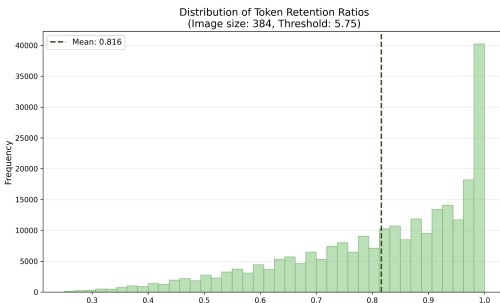

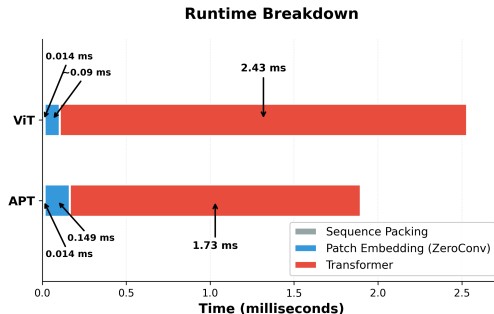

Figure 5: **Sequence Length Distribution.** The distribution of sequence lengths is concentrated near the maximum, and slowly tails off, stopping at around 30% of the maximum value.

Figure 6: **Runtime Breakdown.** Although the standard ViT has a faster patch embedding step, APT's transformer cost is less from the reduced token length, leading to a significant overall speedup.

| Res/Patch | Base (Img/s) | $APT_{\tau=-1}$ |
|---|---|---|
| ViT-B 224/16 | 3310 | 3090 |
| ViT-B 384/16 | 1151 | 1030 |
| ViT-L 224/16 | 883 | 811 |
| ViT-L 336/14 | 395 | 360 |
| ViT-H 224/14 | 441 | 418 |
| ViT-H 336/14 | 190 | 180 |

Table 6: **APT overhead with no reduction.** With no token reduction, APT incurs nontrivial overhead. However, token reduction gives 20%+ speedups relative to the standard implementation, more than covering the discrepancy.

| Model | w/o training | w/ training |
|---|---|---|
| Base | 88.15 | 88.15 |
| Residual | 87.40 | 87.52 |
| NonZero | 87.50 | 87.81 |
| **Zero (Ours)** | 87.98 | 88.13 |

Table 7: **Ablating Zero-initialization.** Using a zero-initialized connection works the best for training APT networks to properly incorporate higher resolution details from the original image, while preserving good-quality predictions before training.

about 10% slower. Next, Figure Figure 6 measures the speed of the ZeroConv and sequence packing operations. When setting $\tau = -1$, the ZeroConv operation is much faster, since no additional computation is incurred. However, when $\tau = 5.75$, it takes about 10% of the end-to-end model latency. The resulting speedup from removing extra tokens more than makes up for this, though, yielding a net 33% speedup. We also perform the entropy computation on the CPU dataloader, parallelizing it across multiple cores and overlapping it with the GPU model computation. This yields no additional overhead.

**Zero-initialization.** Finally, we ablate the use of our *zero-initialized* connection for incorporating higher-resolution details in larger patches. In Table 7, we compare with a simple residual connection, a non-zero initialized connection, and resizing. We find that initializing to zero offers the best off-the-shelf accuracy as well as the strongest performance after an epoch of training. This matches the finding of ControlNet (Zhang et al., 2023a), which showed that zero-initialization works well for adding new capabilities without adding harmful noise to the original model.

**Thresholds.** The main tunable parameter in APT is the entropy threshold, which can differ per scale and controls how compressible a region must be in order to be retained. For most tasks, we use $\tau_3 2 = 5.75, \tau_6 4 = 4.0$, which works well out-of-the-box. We found that object detection required a lower threshold of 2, likely due to the task's reliance on precise edge localization. Despite the smaller threshold, we still observe large speedups in object detection, due to the extremely high resolution inputs. The speed–accuracy trade-off resulting from threshold adjustment is shown in Figure 4, and additional analysis and visualizations are provided in Appendix C. We observe that as the threshold is increased, the accuracy begins to slowly decrease, and eventually the accuracy drops significantly as more and more useful information is blurred in the input.

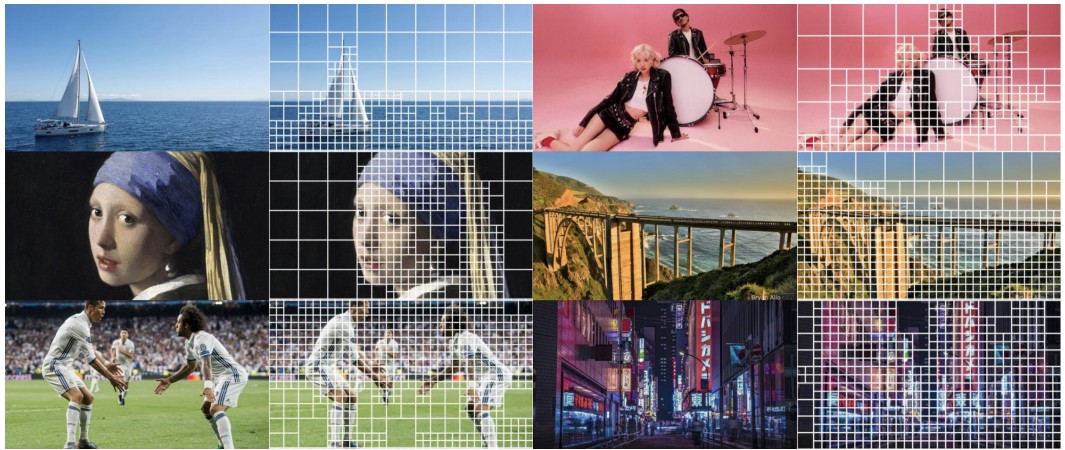

Figure 7: **Visualized Examples.** APT consistently places large patches on more homogenous regions and smaller patches on more complex ones. We use conservative thresholds to limit information loss. Images are best viewed zoomed in. More visualizations are in Appendix C.

## 4.5 EXAMPLE VISUALIZATIONS

We provide qualitative visualizations of the patchification produced by APT with 3 patch scales in Figure 7. As desired, APT consistently assigned larger patches to more homogeneous regions of the image. Dark backgrounds, blue sky, and blurry backdrops are all covered by the largest ($64{\times}64$) patches, while smaller regions that are still simple are given the second largest ($32{\times}32$). Regions with more detail or high frequency receive smaller patches: people's faces or objects in focus are allocated the most. Each image has a different number of patches, depending on its inherent complexity—the cityscape in the bottom right has significantly more than the simpler cartoon image in the top left. By design, the patches produced by APT are agnostic to downstream goals and unaware of what a user might desire from an image. If the top right image were input to a VLM along with the question "What color is the background?", APT would still assign extremely coarse patches to the pink wall due to its textural simplicity. Additional visualizations are provided in Appendix C.

## 5 CONCLUSION

We presented Adaptive Patch Transformer (APT), a method to accelerate ViTs that uses larger patches in simpler areas and smaller patches in more complex ones. It significantly improves training and inference speeds, especially for larger models and higher resolutions. APT can be applied to any pretrained ViT backbone and converges in 1 epoch or less, enabling users to quickly train their models to be faster on a wide range of vision tasks. Our results suggest that APT will benefit the broader vision community by reducing the compute budget required to train state-of-the-art models.

**Limitations.** Although APT provides significant speedups, it still relies on a hand-crafted heuristic to determine patch sizes, which may not always align with downstream users' preferences and could likely be improved. Additionally, APT relies on an empirically-tuned threshold hyperparameter, which can add friction to adoption on downstream tasks. Finally, while APT works for image understanding tasks, it currently does not support image generation, which operates with extremely high-resolution images and large models, making it an ideal application for our work. Future work will be required to overcome these limitations, and we hope that APT can inspire further research on efficient ViTs.

ACKNOWLEDGMENTS

RC is supported by the NSF Graduate Research Fellowship (GRFP). JK is supported by an IITP grant from the Korean government (MSIT) under the AI Excellence Global Innovative Leader Education Program (RS-2022-00143911).

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

## A    IMPLEMENTATION DETAILS

**Hardware Setup.**    All ImageNet experiments were conducted on a node of 8x NVIDIA A100s, and the experiments on object detection, segmentation, and visual QA were conducted with 8xN-VIDIA RTX A6000. The inference-time results were computed on a single GPU, along with the throughput and FLOPS analysis. We used a single node for all work on this paper.

**Layer-level Merging Baselines.**    We used the official repositories for EViT (Liang et al., 2022a), ToMe (Bolya et al., 2022), and DTEM (Lee & Hong, 2024). Since implementations and experiments for ViT-L and ViT-H were not provided, we extended the code to include these two model configurations. In addition to adding the ViT-L and ViT-H variants, all experimental settings (training schedule, enhancements, optimizers, input resolutions, and other hyperparameters) were kept identical to the original baselines to ensure a fair comparison.

**Image Classification.**    We implemented image classification models using the `timm` library (Wightman, 2019), leveraging its pretrained checkpoints. The ImageNet-1K dataset (Deng et al., 2009) was used for training and evaluation, following prior work (Havtorn et al., 2023; Ronen et al., 2023). For the full fine-tuning experiment, we follow the exact MAE training recipe (He et al., 2021), training VIT-B for 100 epochs and VIT-L for 50. We use a base learning rate of 1.5e-3 and use standard augmentations, namely RandAug (Cubuk et al., 2020), Random Erasing (Zhong et al., 2020), random flipping, and cropping. All training was done with 8 GPUs and used batch size 1024. We set layer decay to 0.75 during long fine-tuning. For short fine-tuning, we train the network for 1 epoch with layer decay set to 0.99, and learning rate set to 1e-6, and disable augmentations.

**Visual QA.**    For our Visual Question Answering (VQA) experiments, we utilized the official LLaVA-1.5 (Liu et al., 2024a) implementation and its pretrained checkpoints. Unlike the original approach, which collects data and fine-tunes the entire dataset for one epoch, we fine-tuned only 5% of the dataset, as we initialized from an already fine-tuned checkpoint. To adapt to this setting, we reduced the learning rate by a factor of 10 while following all other fine-tuning procedures recommended by LLaVA. The base image resolution was set to 336 with a patch size of 14, as specified in LLaVA's default configuration. A threshold of 5.75 was applied to determine a patch size of 28.

**Object Detection.**    For object detection, we used the official implementation of EVA-02 (Fang et al., 2024) along with its pretrained checkpoints, which utilize a window attention mechanism. Fine-tuning was conducted following the recommended procedures outlined in EVA-02. Consistent with our previous experiments, we fine-tuned for 5% of the total iterations while reducing the learning rate by a factor of 10. Following EVA-02's settings, the image resolution was 1536 with a patch size of 16. Patch sizes of 128, 64, and 32 were determined based on threshold values of 0.3, 2, and 2, respectively.

**Semantic Segmentation.**    We also utilized the official EVA-02 implementation along with its pretrained checkpoints for semantic segmentation. The ADE20K dataset (Zhou et al., 2019; 2017) was used for training and evaluation. Fine-tuning followed the recommended procedures outlined in EVA-02. In alignment with our previous experiments, we fine-tuned for 5% of the total iterations while reducing the learning rate by a factor of 10. According to EVA-02's settings, the image resolution was either 512 or 640, with a patch size of 16. A threshold of 5.75 was applied.

**Advanced baselines with FlashAttention.**    Many adaptive-token baselines including EViT (Liang et al., 2022a), ToMe (Bolya et al., 2022), and DTEM (Lee & Hong, 2024) incorporate token-weighted attention mechanisms on top of standard scaled dot-product attention. While this weighting is central to their design, it also prevents the use of FlashAttention, forcing these models to rely on the much slower unfused vanilla attention kernel. As a result, the baselines run substantially slower than a standard ViT equipped with FlashAttention, creating an unfair comparison in throughput-oriented evaluations. To address this, we re-implement "advanced" versions of these baselines using the unified operator in Listing 1. During inference, we disable the weighting operation, allowing the model to employ FlashAttention and achieve speeds comparable to modern ViT implementations. This design ensures that all baselines benefit from FlashAttention when possible, providing a more equitable and technically up-to-date comparison across methods.

Listing 1: **Advanced baselines with FlashAttention.** Baselines originally rely on vanilla scaled dot-product attention combined with token-weighted attention, which prevents the use of FlashAttention and significantly slows inference. Our implementation enables a fairer comparison by supporting FlashAttention.

```python
class AdvancedAttention(Attention):
    def forward(self, x, mode="weighted_attn", size=None):
        """
        Args:
            x: Tensor of shape (B, N, C)
            mode:
                - "flash_attn": use torch.scaled_dot_product_attention
                - "weighted_attn": proportion attention with token
                    weights `size`
                - "vanilla_attn": standard scaled dot-product attention
            size:
                Tensor of shape (B, N) with per-token weights.
        """
        B, N, C = x.shape
        out1, out2 = self.qkv(x)
        qkv = out1.reshape(B, N, 3, self.num_heads, self.head_dim).
            permute(2, 0, 3, 1, 4)
        q, k, v = qkv.unbind(0)
        q, k = self.q_norm(q), self.k_norm(k)

        q, k, v = q.float(), k.float(), v.float()
        with torch.cuda.amp.autocast(dtype=torch.float32, enabled=True):
            # FlashAttention without weighting
            if mode == "flash_attn":
                _x = F.scaled_dot_product_attention(
                    q, k, v,
                    attn_mask=None,
                    dropout_p=self.attn_drop.p,
                    is_causal=False
                )

            # Original implementation of baselines
            elif mode == "weighted_attn" and size is not None:
                q = q * self.scale
                attn = q @ k.transpose(-2, -1)
                _attn = attn - torch.max(attn, dim=-1, keepdim=True)[0]
                _attn = _attn.exp_() * size[:, None, None, :].type(torch.
                    float32)
                attn = _attn / _attn.sum(dim=-1, keepdim=True)
                attn = self.attn_drop(attn)
                _x = attn @ v

            # Standard scaled dot-product attention
            elif mode == "vanilla_attn":
                q = q * self.scale
                attn = q @ k.transpose(-2, -1)
                attn = attn.softmax(dim=-1)
                attn = self.attn_drop(attn)
                _x = attn @ v

            else:
                raise ValueError(f"Unknown attention mode: {mode}")

        x = _x.type(x.dtype)

        x = x.transpose(1, 2).reshape(B, N, C)
        x = self.proj(x)
        x = self.proj_drop(x)

        return x
```

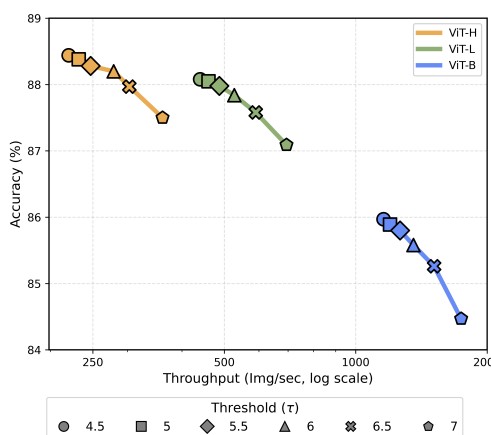 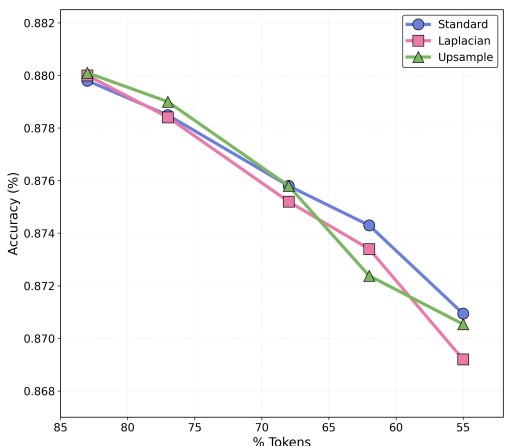

Figure 8: **Threshold Effect.** Increasing the threshold increases throughput significantly, but after approximately $\tau = 6.0$, the accuracy begins to severely drop off, and is not 'fixable' with fine-tuning.

Figure 9: **Analyzing Scorers.** We compare the accuracy on ViT-L/336 for different scorers, controlling for the fraction of retained tokens. We find that the the entropy (standard) scorer performs best at high reductions, but that all three are relatively similar.

## B  RUNTIME DETAILS

All ImageNet experiments were conducted on a node of 8x NVIDIA A100s, and the experiments on object detection, segmentation, and visual QA were conducted with 8xNVIDIA RTX A6000. The inference-time results were computed on a single GPU, along with the throughput and FLOPS analysis. We used a single node for all work on this paper.

## C  ADDITIONAL RESULTS

We provide additional visualizations to illustrate how APT (Adaptive Patch Token) prunes tokens and to analyze the qualitative effects of varying the difference threshold $\tau$, augmentation and scorers. All visualizations were conducted using images at a resolution of $336 \times 336$ and a patch size of $14 \times 14$.

**Threshold Analysis.** The main tunable parameter in APT is the entropy threshold, which can differ per scale and controls how compressible a region must be in order to be retained. Lower values indicate higher sensitivity, and for the vast majority of experiments in this paper, we used $\tau_1 = 5.75, \tau_2 = 4.0$. In Figure 8, we vary $\tau_1$ for 3 model scales with resolution 336 and patch size 14, measuring ImageNet accuracy. We observe that for threshold values larger than 6.0, accuracy drops significantly, while throughput continues to increase. We find that 5.75 offers a good trade-off between acceleration and maintaining quality and hypothesize that this is close to the 'true' threshold for compressibility; beyond this point, coarse-scale patches result in information loss. Figure 10 shows a diverse set of sample images and how our method prunes tokens with relatively lower amounts of information (e.g., background regions or uniform color patches). We fix $\tau_2 = 4.0$ and change $\tau_1$ from 4.5 to 7. Observing various categories of images, one can see that patches containing high-frequency details or salient object features are consistently preserved. In contrast, less critical regions—such as large uniform areas—are pruned. This visualization confirms that the model potentially increases efficiency by ignoring parts of the image that contribute less to the downstream task.

**Scorer Analysis.** Figure 11 qualitatively contrasts the results of an entropy-based scorer with two alternative scores. The entropy-based scorer measures how diverse or complex the distribution of pixel-values within a patch is. If a patch has pixels with a wide range of intensities or colors, it scores higher and is more likely to be retained. This approach naturally favors regions with

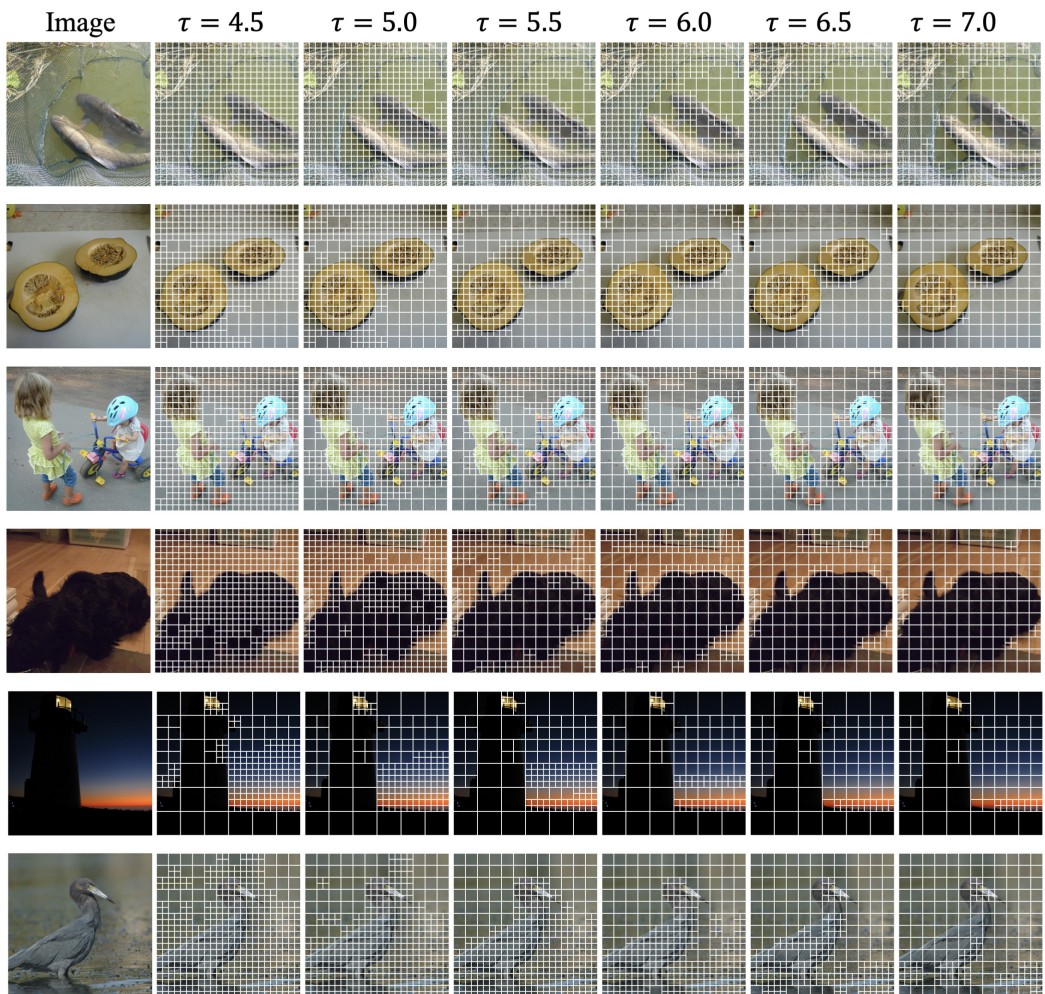

Figure 10: **Threshold visualization.** We can see that patches containing high-frequency details or salient object features are consistently preserved under various thresholds. We used $\tau = 5.5$ for most of the experiments. Zoom in for the best view.

intricate textures, multiple color transitions, or high levels of detail. In comparison, the *Laplacian-based scorer* uses a second-derivative operator (or second-order difference) to detect edges or sharp transitions. Specifically, it looks at how abruptly the pixel intensity changes within a patch. As a result, if there is a strong boundary or a sharp difference in color or brightness, the Laplacian score becomes high, signaling that the patch likely contains important edge information and should be preserved. Finally, we tested an *upsampling-based* scorer, which downsamples the image by a factor of $2^s$ for each scale index $s$, then upsamples back to the original resolution. It then compares the average mean squared difference for each patch. This scorer performs similarly to the Laplacian scorer, but can be a little less sensitive to smaller details.

We also measured the accuracy of using each scorer, controlling for the fraction of reduced tokens, the results of which are shown in Figure 9. Although they perform similarly, the entropy scorer works better at higher token reductions. At higher token reductions, the Laplacian and upsampling-based scorers tend to remove more information that is critical to the model, which results in slightly worse performance. However, the differences are quite small and in practice we expect all three could be used interchangeably.

**Augmentation Analysis.** We compare how APT operates under different data augmentation techniques in Figure 12, since these apply transforms to images that make them 'less natural'. In partic-

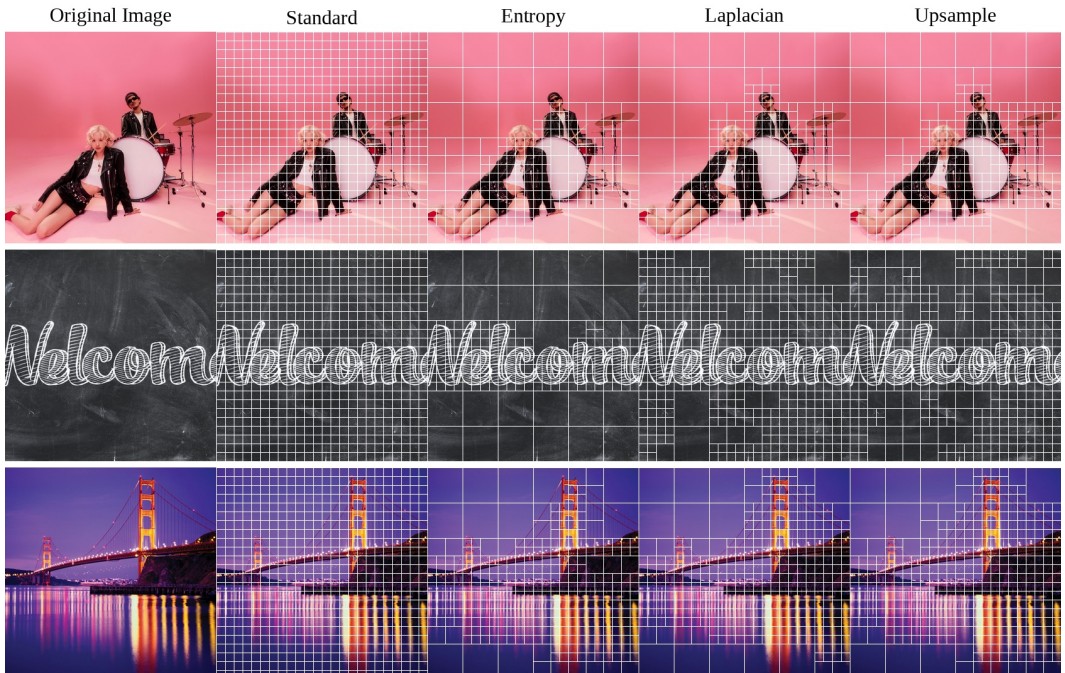

Figure 11: **Scorer visualization.** The entropy, Laplacian and upsampling scorers follow generally the same patterns with minor variations. The entropy scorer uses larger patches on regions with very few differing colors, while the upsampling and Laplacian scorers consistently use small patches on high-texture regions.

ular, random erasing removes parts of the image, causing the overall information to be reduced from the outset. As a result, the total number of retained tokens also decreases because many regions lose their distinguishing features. This phenomenon implies that the speed-up gain could be higher during training or fine-tuning—when augmentations are applied repeatedly—than during inference.

**Qualitative Results.** APT generalizes effectively to downstream visual tasks that require spatial precision, including object detection and semantic segmentation. As illustrated in Figure 13 and Figure 14, APT reliably allocates larger patches to uniform background regions while preserving fine-grained structures with smaller patches around object boundaries and textured areas. Each results support accurate bounding box regression and maintain the pixel-level fidelity necessary for segmentation, demonstrating that APT can deliver significant computational savings without compromising spatial detail or task performance.

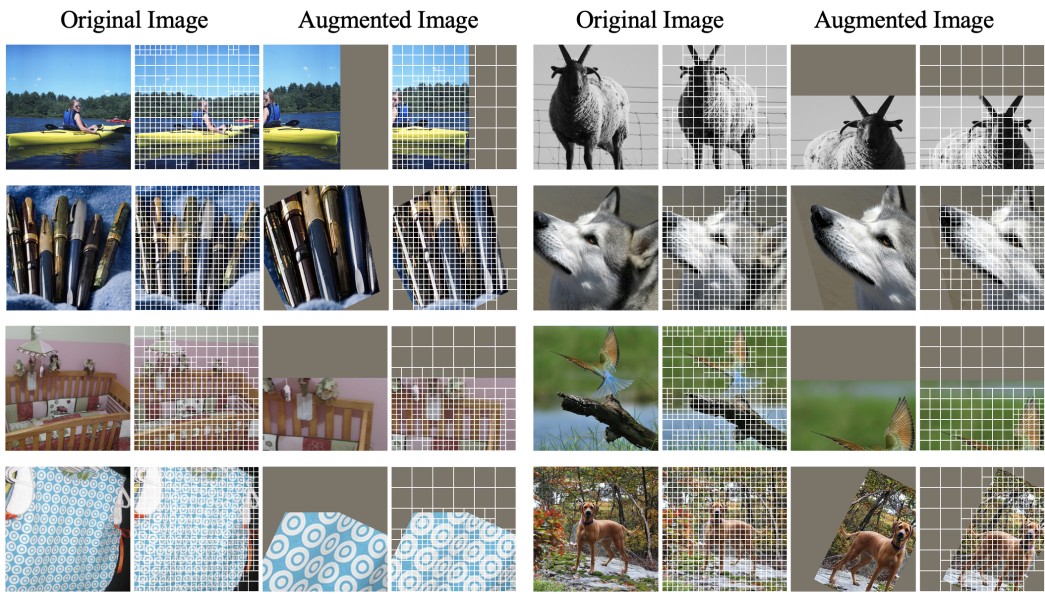

Figure 12: **Augmentation visualization.** We observe that augmentations generally lead to *fewer* tokens. In particular, Random Erasing (Zhong et al., 2020), leads to regions that can be tokenized with the large patch sizes, significantly increasing throughput compared to inference time.

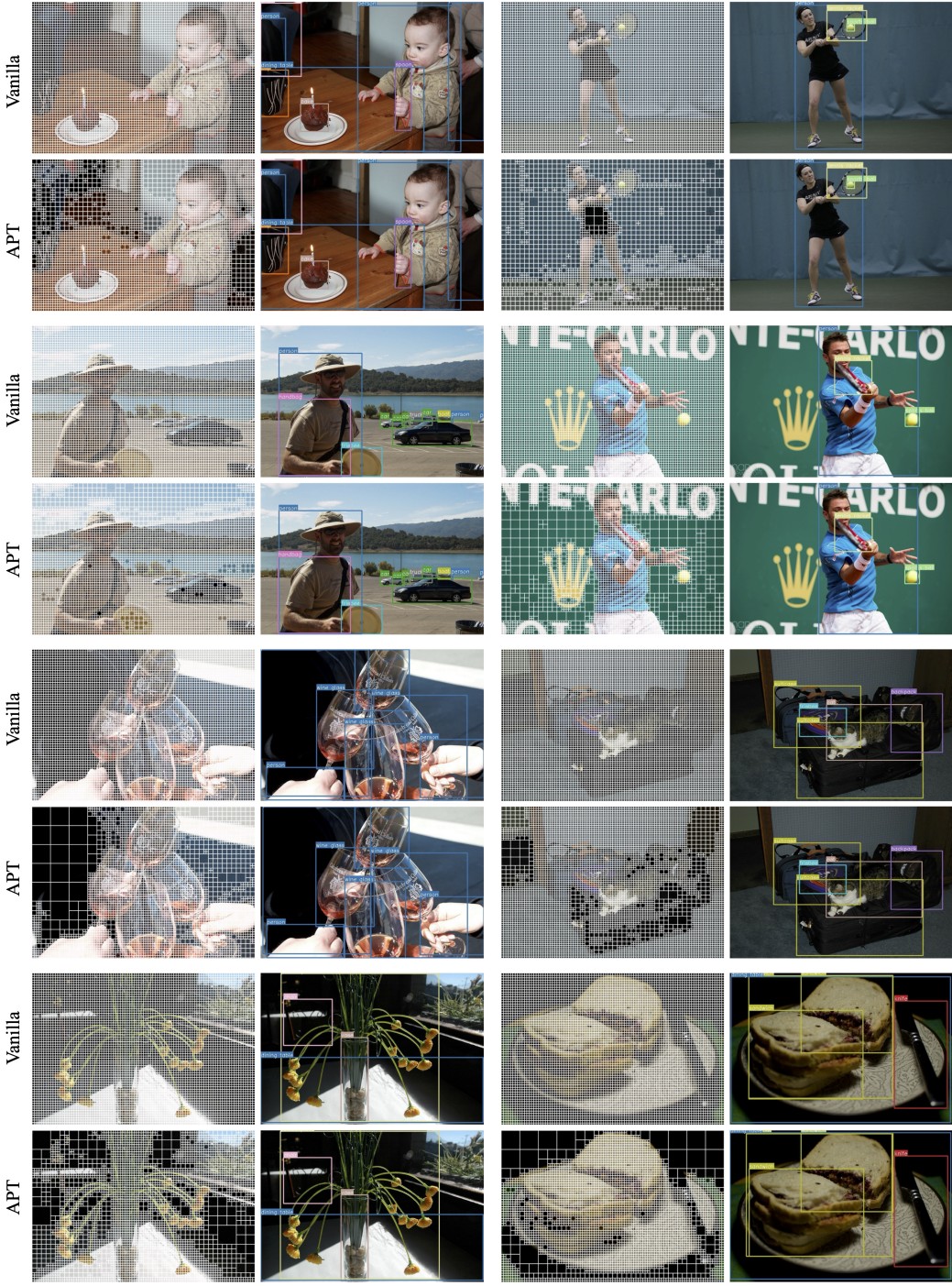

Figure 13: **Object Detection Examples.** First and third columns show the adaptive patch layouts produced by APT, where larger patches correspond to more homogeneous regions and smaller patches capture high-frequency object details. Second and fourth columns show the corresponding object detection outputs, demonstrating that APT preserves essential features for accurate bounding box prediction despite reducing the number of tokens. Images are best viewed zoomed in.

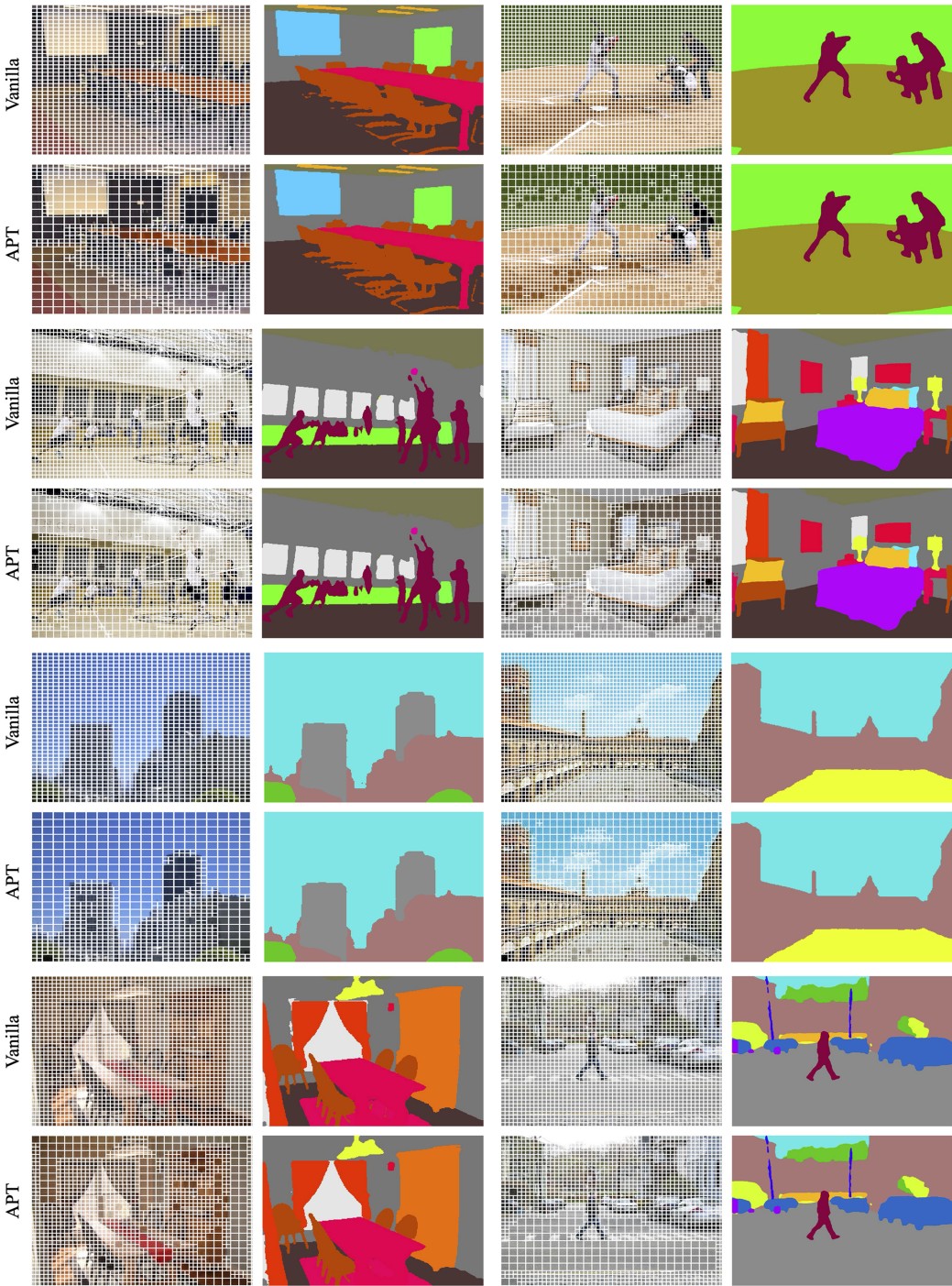

Figure 14: **Semantic Segmentation Examples.** Left and third columns visualize the adaptive patch assignments generated by APT, illustrating how fine-grained regions (e.g., object boundaries) receive smaller patches. Right and fourth columns display the resulting segmentation maps, showing that pixel-level details are preserved sufficiently for dense prediction tasks, even under token reduction. Images are best viewed zoomed in.

