# OpenReview forum: "Faster Vision Transformers with Adaptive Patches"
_ICLR.cc/2026/Conference — ICLR 2026 Poster_

### Official Review · Reviewer_W17G · 2025-10-25

**Soundness:** 3
**Presentation:** 2
**Contribution:** 2
**Rating:** 4
**Confidence:** 4

**Summary:**

The paper proposes the Adaptive Patch Transformer (APT): it adaptively employs multiple patch sizes within the same image based on content—using larger patches for flat/redundant regions and smaller patches for detail-rich areas, thereby reducing input tokens and boosting throughput. Core Implementation: Multi-scale histogram entropy (Eq.(1)) serves as compressibility metric, while hierarchical thresholds (ω) determine whether to “stop” at a layer or continue subdivision. For large patches, dual-path information is simultaneously utilized: “Embedding resized to base dimensions + sub-patch embeddings aggregated via Conv,” fused through a zero-initialized MLP. Inference/training side: “Sequence packing + block-diagonal mask” accommodates variable sequence lengths, accelerated by FlashAttention. Experiments show: Full ImageNet and 1-epoch fine-tuning achieve 20%–90% throughput gains while maintaining accuracy, with accelerated performance on LLaVA's VQA, COCO detection, and ADE20K segmentation. Authors also report: Without token reduction, APT incurs non-zero overhead; zero-initialized connections deliver the most stable “plug-and-play” fine-tuning convergence.

**Strengths:**

1. Comparison tables are provided across four task categories—classification, VQA, detection, and segmentation—clearly demonstrating how variable tokens are transformed into regular feature maps for dense prediction tasks.

2. Key methodological formulas and structural diagrams are presented, along with component ablation studies (zero initialization vs. non-zero/residual; system overhead without compression) to facilitate reproducibility and pinpoint sources of improvement.

**Weaknesses:**

1. The adaptive patch size relies on a hierarchical entropy threshold (Eq. 1, §3.1), which is fixed and manually tuned for each scale. The paper gives no data-driven method to set these thresholds, and poor choices can cause information loss or accuracy drops.

2. To use FlashAttention, some baselines were re-implemented with modifications like disabling weighted attention (§4.1). These changes may alter results, so runtime and throughput comparisons might not be fair without full implementation details or code release.

3. The reconstruction method (§3.3) repeats large-patch features to form dense grids, which may create block artifacts and hurt small-object accuracy. The paper reports only overall mAP/mIoU and shows no failure examples, leaving fine-detail loss unverified.

4. Table 6 shows that APT is slower than the baseline when no compression is applied, suggesting preprocessing overhead from entropy computation and token packing. The paper omits CPU/GPU timing breakdowns, and speed gains are smaller than FLOPs reductions, implying memory or pipeline bottlenecks.

5. The method mainly benefits high-resolution, large-model settings. Key parameters (thresholds ω, binning strategy, search range) are missing, and code is unreleased, making reproduction and fair comparison difficult.

**Questions:**

1. How are the hierarchical thresholds (ωᵢ) determined for different datasets or tasks? Are they manually tuned or selected automatically?

2. How is the histogram entropy computed — what bin settings and value ranges are used? Have other texture or frequency-based criteria been compared?

3. What is the preprocessing overhead of multi-scale entropy computation and token packing on CPU and GPU? Are these times included in the reported runtime measurements?

4. Since token counts vary across samples, what is the distribution of sequence lengths, and how does this variation affect throughput and latency?

5. Can the object detection results be further analyzed by object size (small, medium, large) to better understand performance on fine-grained details?

6. Does the “repeat 2^{2i}” reconstruction step cause aliasing or checkerboard artifacts? Has any boundary or contour accuracy been evaluated to confirm visual quality?

7. How are positional embeddings handled for variable patch sizes and packed sequences? Is there any scale-aware interpolation or adjustment applied?

8. What is the parameter and memory overhead introduced by the zero-initialized MLP fusion layer, and how stable is it during longer fine-tuning?

9. For baselines adapted to support FlashAttention, can the authors provide a detailed list of implementation changes and verify that all models were tested under identical conditions?

10. Does the aggressive merging of smooth regions lead to errors when fine-grained or background details are required for prediction? Has any failure case analysis been performed?

**If you address my concerns, I will consider raising my score.**

---

> ### Author Response · Authors · 2025-11-24
> **Response: Reviewer W17G**
>
> We thank the reviewer for their thorough review, as well as the comprehensive feedback. We address their outlined concerns below.
>
> __Weakness: Reliance on Threshold__
>
> The reviewer is correct that our threshold is a hyperparameter that requires tuning. However, we provide an empirical measurement of the accuracy-efficiency tradeoff from the threshold in Figure 6, which helps understand how threshold values empirically affect performance.  Furthermore, we found that our choice of thresholds (5.75 and 4.0) generalizes well across most tasks, and we discuss this on Lines 954-967 of the Appendix. Finally, we include an interactive tool to understand how hyperparameters influence the patches on various input images on our anonymous project page (https://clamsoup97.github.io/anonymous-projects/apt/).  We highly encourage reviewers to visit the page to better understand APT’s threshold and its effect. While we agree that tuning the thresholds is not ideal, usage of hyperparameters is common throughout computer vision; we leave addressing this limitation to future work, and have updated the Limitations section accordingly. We do not believe that this is a core weakness of APT or that it justifies rejecting our work.
>
> __Weakness: Details on FlashAttention version of baselines.__
>
> We agree that these details are crucial to explain thoroughly, and have updated the Appendix to comprehensively list all modifications on Lines 850-859. In general, baselines that rely on attention scores are totally incompatible with FlashAttention. Those that do not rely on scores typically use some sort of weighting mechanism, such as Token Merging or DTEM. Additional weighting is incompatible with FlashAttention, but disabling this allows us to give a more fair comparison to these method’s speed. In Figure 4, we implement both the original version (shown in Figure 4) and a version that disables weighting and uses FlashAttention, which we refer to as `advanced’. FlashAttention consistently speeds up these baselines at the cost of a small accuracy drop - we believe that this is a true limitation for these methods compared to APT, which is natively compatible with hardware-aware kernels due to its simplicity.
>
> __Weakness: Object Detection: Artifacts on Dense Feature Maps, Failure Cases and Further Metrics.__
>
> We emphasize that our paper does not claim to be a state-of-the-art method for object detection; rather, we aim to match performance while achieve a speedup. The reviewer is correct that repeating and upsampling the output features to produce a dense feature map does produce artifacts. However, this does not affect overall performance significantly. The large patches tend to be assigned in smooth regions, where objects and edges rarely occur; artifacts in the feature map here do not affect performance whatsoever. However, we have included a failure case in our project page where this does make a difference: when an object boundary is close to the background in intensity, it is possible to apply large patches to edges, as shown on the toilet in the image. However, we observe empirically that this does not affect overall mAP, and our method achieves a strong speed-up as well.
>
> We next measure the object detection metrics on different object scales.
>
> |          | AP    | AP50  | AP75  | APs   | APm   | API   |
> |----------|-------|-------|-------|-------|-------|-------|
> | Vanilla  | 62.28 | 80.80 | 68.10 | 45.85 | 66.75 | 78.01 |
> | Resizing | 61.75 | 80.27 | 67.46 | 45.35 | 66.41 | 77.05 |
> | Ours     | 62.07 | 80.64 | 67.90 | 45.52 | 66.55 | 77.25 |
>
> APs empirically exhibits the smallest deviation compared to APm and APl. This is because the entropy-based patchification strategy reliably assigns the smallest patches to high-frequency regions—where small objects predominantly appear—thereby preserving critical fine-grained details. In contrast, medium and large objects contain broader and more homogeneous areas, which are more likely to be merged into larger patches, leading to comparatively larger fluctuations in APm and APl. Importantly, these changes do not necessarily indicate a structural weakness in recognizing large objects; rather, they appear to stem from the intrinsic evaluation characteristics of IoU for large objects. Since large objects occupy a substantial spatial extent, even minor boundary imperfections can produce disproportionately larger changes in IoU, amplifying the apparent variation in APl. As a result, the observed differences in APl are more reflective of the metric’s sensitivity than of any fundamental degradation in the model’s ability to localize large objects.

---

> > ### Author Response · Authors · 2025-11-24
> > **Response: Reviewer W17G (2/3)**
> >
> > __Weakness: Overhead to packing and lack of runtime details.__
> >
> > There is indeed a small overhead to packing and re-arranging the tokens, since this must be done dynamically based on the input content, rather than the input size only as is common for language. We have measured this in detail in Figure 6 and Table 6, both of which demonstrate that the overall model speedup is significantly more than the cost of the additional overhead. While Reviewer W17G is correct that the speedup is less than the FLOPs reduction, this is extremely common in efficiency works, especially for attention. Several works that reduce the cost of attention or number of tokens report speedups that are closer to linear in the token reduction rather than quadratic, as theoretical FLOPs are. For example, until FlashAttention [1], most efficient-attention work achieved wall-clock time nowhere close to the theoretical FLOPs reduction. Since this observation is common in the literature, we disagree that this is a weakness of our work.
> >
> > __Weakness: The method mostly benefits large models and high resolutions, and code is missing.__
> >
> > Our method achieves improvements on models range from ViT-B to ViT-H, from resolutions as low as 224 to as high as 448. While APT yields more speedup on larger models and resolutions, it is inaccurate to say that it does not benefit lower resolutions, and the models that we test are all small enough to fit on a single A100 GPU. We also explain which parameters are used in the Appendix, and have released an anonymized version of the code (https://github.com/clamsoup97/apt), which is also available on our project page. We encourage reviewers to check the code in order to better understand what design choices were made.
> >
> > __Question: How are thresholds for different tasks decided?__
> >
> > We decided the threshold manually for different tasks, but found that across most tasks the same thresholds worked well. The only exception was object detection, which required lowering the threshold. As stated above, we do not believe that tuning a single hyperparameter constitutes a core weakness of our idea, and we believe future work can address learning an adaptive threshold to improve upon APT.
> >
> > __Question: How is entropy computed, and have other scorers been tested?__
> >
> > For computing entropy, we use 512 bins for finer granularity and the input value range is (0, 255) since we operate on RGB inputs. This computation is done in `compute_patch_entropy_vectorized' in the file entropy_utils.py in our anonymized code release. We also included an analysis of alternative scorers beside entropy in the Appendix: we test using a Laplacian filter as well as an upsampling-based metric. We find that these metrics perform comparably, but slightly worse than entropy. While the analysis with the Laplacian was included in our initial submission’s supplement, we have updated it to also include the upsampling scorer as well, along with a Pareto plot to measure accuracy vs fraction of reduced tokens.
> >
> > __Question: Preprocessing overhead of entropy and packing.__
> >
> > The overhead is included in all reported runtime measurements in the paper, since we measure end-to-end latency. The entropy computation is done on the CPU data loader, and is entirely overlapped with the model inference; it adds no overhead, even though we are doing extra computation.  The token-packing step is extremely fast, and constitutes about 2-3% of the model latency. Additional overhead comes from the ZeroConv layers in the patch embedding, which constitutes about 16% of the APT latency; however, the decrease in sequence length more than makes up for this, as shown in Figure 6.
> >
> > __Question: Distribution of sequence lengths.__
> >
> > This is a great point. We have included a plot in the Appendix of the distribution of sequence lengths in ImageNet at our chosen thresholds, as well as a plot of total throughput as a function of sequence length. The variance in sequence length does affect throughput, as we demonstrate in Table 1. The `Random’ baseline uses the mean sequence reduction (18%) for every input, and randomly removes that fraction of tokens. The fact that this is faster than APT shows that this variance does lead to a small overhead. However, because our size is much smaller on average than vanilla ViT, we still achieve a significant speedup.
> >
> > __Question: Dense feature map visual quality.__
> >
> > We have provided a failure case on our project page. However, we emphasize that the goal of our paper is not to produce high visual-quality feature maps; it is to achieve as large a speed-up as possible without hurting performance. Although feature maps may have artifacts, if this does not affect downstream overall performance, we do not believe this is a core weakness of the paper.

---

> > > ### Author Response · Authors · 2025-11-24
> > > **Response: Reviewer W17G (3/3)**
> > >
> > > __Question: Positional encodings.__
> > >
> > > We apologize that this was not clear in the main text and have updated the manuscript to include a detailed explanation on Page 5, Lines 238-244. Reviewer W17G is correct that scale-aware interpolation is applied. Specifically, if the base patch size is 16x16, each patch in the HxW image has a positional encoding, resulting in a grid of H/16 x W/16 patches. To get positional encodings for 32x32 patches, we downsample the grid to H/32 x W/32, and apply the resulting positional encodings to any 32x32 patches assigned to the input. Intuitively, this corresponds to sampling positional encodings at the patch centers of the larger patches. This procedure works for sinusoidal, learned and RoPE positional encodings.
> > >
> > > __Question: ZeroConv parameter and memory cost.__
> > >
> > > The ZeroConv parameter count depends on the embedding dimension of the model. For a ViT-L, it adds 5.2M and uses 10MB more memory (in bf16 precision), out of a total of 310M parameters and 620MB for the full model, an increase of 2%. Since it is a single conv layer and the models we train in this paper are small enough for a single 8xA100 node, there are no issues with training stability for longer fine-tuning.
> > >
> > >
> > >
> > > We appreciate your comprehensive feedback! Given that we have addressed your concerns, would you consider raising your score to a positive one?

---

> > > > ### Comment · Reviewer_W17G · 2025-11-24
> > > >
> > > > Thank you for the additional experiments, threshold sensitivity analysis, and implementation details provided in the rebuttal. These additions improve the clarity the method. So, the work is solid and practically useful from an engineering view, especially in large-model and high-resolution settings, where the proposed entropy-based adaptive patchification and token packing achieve around 20–50% real throughput improvement while largely preserving downstream performance.
> > > >
> > > > However, the method still relies on **hand-crafted entropy thresholds** and **heuristic scoring**, without a learning-based or task-aware mechanism for scoring or threshold selection. But, it's ok. Additionly, it lacks quantitative comparison with the most closely related multi-granularity methods (e.g., MG-ViT), leaving its relative advantage within this line of work not entirely clear. From a **model-compression view**, the approach focuses on **token-level** compute reduction rather than **parameter compression**, making it more suitable as a complementary front-end to pruning, quantization, or distillation rather than a independent compression technique. However, I still think that the engineering implementation is done well in this work.

---

> ### Author Response · Authors · 2025-11-25
> **Response: Reviewer W17G**
>
> Thank you so much for your quick response to our prior points! We are glad that you find our work to provide a strong speedup and that our implementation is high-quality, and we genuinely appreciate your willingness to engage in discussion. We have addressed your new concerns below:
>
> __Reliance on heuristics.__
>
> While it would be ideal not to rely on any thresholds or heuristics, we do not believe that this is a disqualifying weakness. Language tokenization methods like BPE [1] or SentencePiece [2] are similar heuristic-based adaptive tokenizers, grouping bytes into useful chunks, and have had an outsized impact in language modeling: they are yet to be supplanted by any learned method. Furthermore, the use of a heuristic for tokenization is extremely fast, easy to integrate with existing vision transformer implementations, and allows APT to be deployed without additional training. We believe that future work can address this limitation, and that tuning a single hyperparameter does not constitute a reason to reject our paper.
>
> __Comparison to other methods (e.g, MG-VIT)__
>
> As we noted in our response to Reviewer sSUC, while MG-VIT does use multiple scales, it reports *extremely limited results* and is *incompatible with FlashAttention*, limiting its speed significantly. Furthermore, MG-VIT has no released code and depends on several complex hyperparameters, making re-implementation challenging and, in our view, of little utility for comparison. We instead compared APT to an improved version of QuadFormer, a multi-patch-size method that outperforms MG-VIT and which we believe is the strongest comparable baseline, as well as the best possible pruning and merging-based methods. We also included a new comparison to PPT [4] as per Reviewer sSUC's request, which further solidifies our method's empirical strength compared to baselines.
>
> __Combination with Pruning and Distillation.__
>
> As we also noted in our response to Reviewer sSUC, it is true that in theory APT should be compatible with pruning-based works.  However, we found empirically that the strongest pruning methods, such as Token Merging, do not provide any improvement when combined with our method. In fact, we observe a small decrease in the accuracy-speed tradeoff when combining pruning with APT! Since APT outperforms the comparable merging and pruning baselines in both speed and accuracy we believe APT is a strictly superior strategy.
>
> APT is certainly compatible with quantization and distillation, and is definitely a token-reduction-based method, not a parameter compression method. However, the fact that orthogonal methods exist for accelerating models does not, in our view, detract in any way from the value of our work. Improved tokenization methods for language models such as BPE [1] or SuperBPE [3] are compatible with quantization and distillation, yet add significant value on their own. If applied to a distilled or quantized transformer, APT would still reduce the input size and lead to large speedups.
>
> Given that we have addressed your concerns, would you consider raising your score to a positive one? Again, we appreciate your feedback and quick response!
>
> [1] Sennrich, R., Haddow, B. and Birch, A., 2015. Neural machine translation of rare words with subword units. arXiv preprint arXiv: 150807909.
>
> [2] Kudo, T. and Richardson, J., 2018. SentencePiece: A simple and language independent subword tokenizer and detokenizer for neural text processing. arXiv preprint arXiv:1808.06226.
>
> [3] Liu, A., Hayase, J., Hofmann, V., Oh, S., Smith, N.A. and Choi, Y., 2025. Superbpe: Space travel for language models. arXiv preprint arXiv:2503.13423.
>
> [4] Wu, Xinjian, et al. "Ppt: Token pruning and pooling for efficient vision transformers." arXiv preprint arXiv:2310.01812 (2023).

---

> > ### Comment · Reviewer_W17G · 2025-11-25
> >
> > Thank you for your latest response. My concern has been resolved, and I will update my rating accordingly.

---

> > > ### Author Response · Authors · 2025-11-28
> > >
> > > Thanks! We appreciate you raising your score to a 6 and your many helpful points of feedback. Your review has helped us significantly improve the paper.

---

### Official Review · Reviewer_u6bT · 2025-10-26

**Soundness:** 3
**Presentation:** 3
**Contribution:** 3
**Rating:** 6
**Confidence:** 4

**Summary:**

This paper introduces an adaptive patching (tokenization) method for ViTs. The main idea is to measure the entropy (how variable the pixel values are) of local regions of images and use finer (smaller) patches to tokenize the regions with higher variation. The patches of different sizes are resized (and split) to the same size for embedding. It reduces the number of tokens compared to the ViT using the same-sized patches for the whole image. This method is adapted to different ViTs and used in different vision tasks such as ImageNet classification, VQA, and object detection. The method generally shows better accuracy-effeciency trade-off than the original ViTs and some efficient ViTs.

**Strengths:**

- This paper is well written overall; the adaptive patching idea (larger patches for low-entropy regions, smaller for high-entropy regions) is intuitive and easy to follow. The entropy formulation and hierarchical quadtree patchification are clearly described, with alternatives noted for the appendix.
- The zero-initialized MLP lets the model incorporate high-res details without hurting initialization, enabling quick convergence from existing ViTs.
- The proposed method APT plugs into several ViT backbones and tasks, including classification, VQA, detection, and segmentation. It also works with window attention (e.g., EVA/ViTDet).
- APT reported 40–50% throughput gains on large models/resolutions while matching accuracy, and also some speedups on dense tasks, achieving better accuracy-effeciency trade-off than several well-known efficient ViTs such as EViT and ToMe.
- The paper re-implements layer-level merging baselines with FlashAttention for a fairer comparison (and shows APT outperforms across compute budgets).

**Weaknesses:**

- Table 3 shows +22–26% throughput on LLaVA-1.5 (7B/13B), with some metrics slightly down (e.g., VQAv2 −0.6 for 13B) and others on par or up; overall the Pareto looks close but not strictly better across all benchmarks. Table 4 shows +14–30% throughput on detection with essentially unchanged mAP/AP50. This is positive, but the improvements are less decisive than in classification and would benefit from a Pareto plot analogous to Fig. 4 for these tasks.
- Currently APT uses entropy to measure the variation of pixels in image regions. It would be beneficial to add ablations for different measures, such as standard deviation of the pixels and local frequency (e.g., DCT-band energy).
- The writing of the experimental setup can be clearer, specifically on the difference between Full Fine-Tuning and Short Fine-Tuning (Section 4.2). Sometime it could be a bit confusing as to what is the pre-trained MAE; is it one only trained with masked autoencoding or one with both masked autoencoding and classification training?

**Questions:**

- For dynamic input size (seciton 3.3), you concatenate the tokens of a batch of images into a single sequence and use block attention. Why not padding the sequence of each image into the same length?
- In the Input-level Merging Baselines, what do you mean by saying Resizing represents a stronger version of Quadformer? It seems that Resizing is a variant of APT by removing the zero initialized layer, and is not really related to Quadformer. Similar question for "Random". Clearer explantion is needed.
- For the dense prediction tasks (section 4.3), you mentioned training only the newly added component. Does it mean only training the conv layers and ZeroMLP as in Fig 3? For example, for LLaVa, the language model and the projection layer are frozen. Is that correct?

---

> ### Author Response · Authors · 2025-11-24
> **Response: Reviewer u6BT**
>
> We are glad that the reviewer appreciated our paper's writing quality as well as its strong efficiency improvements. Below, we address the concerns they outlined.
>
> __Weakness: additional Pareto curve for other visual tasks.__
>
> This is an excellent suggestion, and we will gladly add this Pareto curve to the next version of the paper. We agree that the improvement is more modest in the VQA and dense prediction tasks; the idea with these experiments was to demonstrate that APT still provides a speed-up without harming performance even on tasks that require strong pixel-level understanding. Fine-tuning several models at different compute budgets to properly measure the compute-performance tradeoff will take some time, however, and we will do our best to add it here as soon as the experiment is complete.
>
> __Weakness: Ablating alternative scorers.__
>
> Thank you for this feedback. We measured the effect of APT with two other scores (Laplacian and Upsampling) in the Appendix, and show that they perform quite similarly, with entropy having a slight edge. We also visualize their differences in Figure 9 of the Appendix, and measure their efficiency-accuracy tradeoff in Figure 7.  We included the Laplacian analysis in the initial submission supplement, but have updated the text to also include the Upsampling scorer, both of which are explained in Lines 960-1020 on Pages 18 and 19.
>
> __Weakness: Clarifying Experimental Setup.__
>
> Thank you for pointing this out, and apologize for the lack of clarity. We have updated the text in Lines 311-322 on Page 6 to make the difference between these clearer.  Here, “pre-trained MAE” means the backbone trained with masked autoencoding only. The full-fine tuning experiments measure APT’s performance after training for 50+ epochs from a self-supervised pretrained ViT backbone, i.e the standard ImageNet fine-tuning pipeline. The “short fine-tuning experiments” apply APT to an already fine-tuned image classification network, then fine-tune for one epoch to train the ZeroMLP layers. The ZeroMLP layers do not negatively affect performance since they are initialized to zero.
>
>
> __Question: Why not padding?__
>
> Current state-of-the-art transformers typically do not use padding to handle variable-size sequences since FlashAttention natively supports packing, and applying padding makes the entire operation slower: either a complex attention mask needs to be constructed, or the model pays attention to padding tokens, which is unnecessary additional compute [1, 2, 3]. Padding would also require every sequence to be padded to the length of the maximum-size input in the batch, which significantly reduces APT’s memory savings.
>
> 1. Dao, T., 2023. Flashattention-2: Faster attention with better parallelism and work partitioning. arXiv preprint arXiv:2307.08691.
> 2. Dehghani, M., Mustafa, B., Djolonga, J., Heek, J., Minderer, M., Caron, M., Steiner, A., Puigcerver, J., Geirhos, R., Alabdulmohsin, I.M. and Oliver, A., 2023. Patch n’pack: Navit, a vision transformer for any aspect ratio and resolution. Advances in Neural Information Processing Systems, 36, pp.2252-2274.
> 3. Krell, M.M., Kosec, M., Perez, S.P. and Fitzgibbon, A., 2021. Efficient sequence packing without cross-contamination: Accelerating large language models without impacting performance. arXiv preprint arXiv:2107.02027.
>
> __Question: Clarifying Quadformer vs Resizing baseline.__
>
> We apologize that our point was not clear. For any input image, QuadFormer applies a constant number of larger 32x32 patches with a top-k heuristic. These 32x32 patches are then simply resized to 16x16, and used as part of attention. Our baseline is a stronger version because it uses a stronger heuristic, and does not use a constant number of patches, adaptively selecting how many patches to replace with larger patch sizes. We have updated the text to make this clearer on Lines 265-268 of Page 5 with changes marked in blue.
>
> __Question: How are dense tasks fine-tuned?__
>
> We apologize for the earlier confusion. We clarify our fine-tuning strategy as follows: For detection and segmentation, we follow the same full fine-tuning protocol used in our ImageNet classification experiments. That is, we fully fine-tuned the entire backbone together with APT, using layer-decay. This ensures a fair comparison with the baseline models, which are also fully fine-tuned under the same recipe. In contrast, for VQA the underlying multimodal model is extremely large, making full fine-tuning computationally prohibitive. Therefore, we froze the original vision backbone and language model, and fine-tuned only the newly added APT modules. This lightweight adaptation still allowed APT to recover or exceed baseline performance while providing speedups. We have updated the manuscript to clearly reflect these two different training regimes.
>
>
> If we have sufficiently addressed your concerns, we humbly request that you consider raising your score, and are happy to answer any further questions.

---

### Official Review · Reviewer_sSUC · 2025-10-29

**Soundness:** 2
**Presentation:** 3
**Contribution:** 3
**Rating:** 2
**Confidence:** 5

**Summary:**

This paper proposes Adaptive Patch Transformers (APT) , which accelerates Vision Transformers (ViTs) by replacing uniform patch splitting with content-aware, multi-granularity patching   based on entropy calculation . The method utilizes a Resize + ZeroMLP mechanism to fuse features from different scales into a unified embedding space, significantly reducing the input token count . The key contributions include achieving drastic throughput speedup (up to 50% on ViT-H) while maintaining accuracy across classification and dense prediction tasks, and ensuring   fast, stable adaptation to fine-tuned models via its zero-initialized fusion layer.

**Strengths:**

1. Strong Experimental Validation: The paper features a comprehensive set of ablation studies   across various tasks, effectively demonstrating the efficacy of the proposed mechanisms.
2. Significant Efficiency and Generalization: APT delivers substantial throughput improvements (up to 50% on ViT-H) on large models, exhibits fast convergence (1 epoch fine-tuning), and shows   robust generalization   across classification and dense prediction benchmarks.
3. Clarity and Presentation: The paper is well-structured, and the figures are high-quality.

**Weaknesses:**

1. Missing Similar Methods Comparison: The evaluation is incomplete because it fails to include a direct comparison against methods addressing the same task, such as MG-ViT[1] and PPT[2]. A detailed analysis of the methodological and empirical differences among APT, MG-ViT and PPT would substantially strengthen the paper.

2. Hyperparameter Dependency: The performance is sensitive to the entropy threshold , which appears to be a manually tuned hyperparameter . This dependency might complicate achieving optimal efficiency across different downstream tasks, as the definition of "salient information" can vary significantly between tasks.

3. In object detection tasks, does the use of entropy to determine patch size risk ignoring subtle object boundaries? Entropy measures pixel intensity distribution diversity, which may not perfectly align with semantically critical edges, especially when compared to gradient-based measures.

4. For higher resolution images, which naturally result in a larger number of base patches, could the authors explore further patch fusion/aggregation operations after the initial adaptive patching step, particularly when several adjacent low-entropy patches exhibit similar entropy scores

5. When patch size change, should entropy threshold change? That means one size patch correspond to one entropy threshold, and different size patch correspond to different entropy threshold.

[1]Zhang Y, Liu Y, Miao D, et al. MG-ViT: a multi-granularity method for compact and efficient vision transformers[J]. Advances in Neural Information Processing Systems, 2023, 36: 69328-69347.

[2]Wu, Xinjian, et al. "Ppt: Token pruning and pooling for efficient vision transformers." arXiv preprint arXiv:2310.01812 (2023).

**Questions:**

Refer to weakness. If these concerns are well addressed, I will raise the rating to a positive one.

---

> ### Author Response · Authors · 2025-11-24
> **Response: Reviewer sSUC (1/2)**
>
> Firstly, we thank the reviewer for their extensive and helpful feedback. We are glad they found our experiments to be strong, and our efficiency improvement to be significant. We have incorporated their advice into our updated submission, and respond to their individual concerns here.
>
> __Weakness: Comparison to PPT and MG-VIT.__
>
> Thank you for pointing out these two works - we missed them in our original literature search, and have updated the Related Works Section to include them. MG-VIT is conceptually similar to APT in that it uses multiple patch granularities for vision transformers. However, MG-VIT has three main shortcomings that APT comprehensively addresses. Firstly, it requires training from scratch (300 epochs for ViT-S). APT can be run without training and can also be fine-tuned for peak performance in as little as 1 epoch, making it much easier to use.  Secondly, MG-ViT requires materializing the attention mask to incorporate attention scores into the method. This prevents it from being able to use FlashAttention, and thus MG-VIT would be at best 20-30% slower than APT.  MG-ViT also only supports two levels of patch sizes and is very limited in scale (only evaluating on ViT-S). Finally, MG-VIT reports no model speed-up whatsoever, and since MG-VIT does not release any code, we are unable to directly compare their results to ours. That being said, we will gladly mention it in the Related Work, since it is important prior art.
>
> Next, PPT is a method that builds on ToMe by combining token merging and pruning. Like MG-VIT, it relies on attention scores for identifying tokens to prune or pool, and thus cannot incorporate FlashAttention for accelerated attention. We have updated the Pareto curve in Figure 4 to include results from PPT. Although the original paper only reports results on DeIT-S and DeIT-B, we adapted it to run on ViT-L and ViT-H. PPT has several hyperparameters , such as token reduction number, number of pooling layers, and score threshold: we are not certain that we tuned them optimally, but we made our best efforts. On ViT-L, we observe that PPT performs slightly better than Token Merging, but is still considerably slower than APT. On ViT-H, PPT is worse than both, which to us suggests that it is best suited for smaller scale models. We have also updated the Baselines section to include a discussion of PPT as well.
>
> __Weakness: Hyperparameter Dependency.__
>
> This is a reasonable observation: APT is indeed dependent on the threshold hyperparameter. We tuned this empirically in Figure 6 of the Appendix. We generally found that we could use the same default thresholds across most tasks, with the exception of object detection, where we did need to lower the threshold slightly. We have updated the discussion of this on Page 9, Lines 466-471, and also include further discussion in Section C of the Appendix. We have also provided an interactive visualization tool on our anonymized project page, at https://clamsoup97.github.io/anonymous-projects/apt/. In sum, we believe that this is a reasonable limitation given that hyperparameters are commonplace in computer vision methods such as PPT, MG-VIT, ToMe, etc and leave it to future work to remove this dependency in a data-driven manner.
>
>
> __Weakness: Entropy on Edges for Object Detection.__
>
> We acknowledge that entropy is not a perfect surrogate for semantic edges, and visualize a case where this occurs in the "Failure Cases" section of our project page. However, the concern that APT may overlook subtle boundaries does not affect overall performance on object detection, as shown in the Table below.
>
> |          | AP    | AP50  | AP75  | APs   | APm   | API   |
> |----------|-------|-------|-------|-------|-------|-------|
> | Vanilla  | 62.28 | 80.80 | 68.10 | 45.85 | 66.75 | 78.01 |
> | Resizing | 61.75 | 80.27 | 67.46 | 45.35 | 66.41 | 77.05 |
> | Ours     | 62.07 | 80.64 | 67.90 | 45.52 | 66.55 | 77.25 |
>
> These numbers demonstrate that APT retains high-IoU accuracy (AP75) almost identically to the baseline, which directly contradicts the claim that subtle boundaries are being lost. If APT were systematically discarding boundary structure due to entropy, AP75 and small-object APs would be the first metrics to collapse—yet they remain virtually unchanged.
>
> Second, APT does not simply downsample low-entropy regions. As described in Section 3.2, APT preserves high-resolution details through sub-patch embeddings aggregated with Conv2D + a zero-initialized MLP, ensuring that even large patches contribute fine-grained information rather than blurred features. This mechanism explicitly mitigates the risk of losing boundary content.

---

> > ### Author Response · Authors · 2025-11-24
> > **Response: Reviewer sSUC (2/2)**
> >
> > __Weakness: Exploration of combining with fusion and aggregation.__
> >
> > This is a valuable suggestion. We did explore combining APT with a representative ToMe-style token-merging module, as in theory, merging should be compatible with our contribution. However, we did not observe any meaningful improvement in initial experiments. Our interpretation is that APT already removes most redundant visual information before feeding tokens into the backbone. Since ToMe introduces overhead from computing token similarities and clustering operations, the marginal reduction it provides on top of APT does not translate into meaningful end-to-end speedups. Furthermore, combining with merging introduces even more hyperparameters and complexity, making the model harder to debug and tune. We leave it to future work to explore this combination to further improve upon our work.
> >
> > __Weakness: Patch Size vs Entropy Threshold.__
> >
> > We found that using a smaller threshold for larger patch sizes worked well, but is not strictly necessary. In particular, we use 5.75 for 32x32 and 4.0 for 64x64 for training-free experiments and short fine-tuning.  However, we found that in full fine-tuning, we could use 5.75 for both patch scales. For consistency, all experiments in the paper default to 5.75 and 4.0 unless otherwise specified.
> >
> >
> > If we have sufficiently addressed your concerns, we humbly request that you consider raising your score to a positive one. If you have further questions, we are happy to answer them as well.

---

> > > ### Comment · Reviewer_sSUC · 2025-11-25
> > > **Many thanks**
> > >
> > > I thank the authors for their response. My concerns have been effectively addressed. Although I believe this idea has become common in 2025, the work is rigorous and solid. Therefore, I have adjusted my rating to positive and am willing to offer support.

---

> > > > ### Author Response · Authors · 2025-11-28
> > > > **Thank you!**
> > > >
> > > > Thanks! We appreciate you raising your score to a 6 and your willingness to engage in constructive discussion. Your feedback has significantly improved our paper.

---

### Official Review · Reviewer_sF7q · 2025-10-30

**Soundness:** 3
**Presentation:** 3
**Contribution:** 3
**Rating:** 8
**Confidence:** 3

**Summary:**

The manuscript proposes Adaptive Patch Transformers (APT) considering using multiple different patch sizes within the same image processed by a Vision Transformer (ViT). Larger patch sizes are allocated in more homogeneous areas while smaller patches are allocated to more complex ones. The proposed approach accelerates the ViT with 40%.

Entropy is used as a measure of a patch’s compressibility with lower entropy indicates higher redundancy.

Patch aggregation is also employed aggregating embeddings from sub-patches back to size

Token merging approaches is done at input-level. Input-level merging reduces tokens directly from image patches before entering the model. The method is also compared to Layer-level token merging.

**Strengths:**

* A more efficient and adaptive transformer model is proposed, where the adaptation refers to the fact that complex information is processed in more detail with smaller patches. Meanwhile less complex regions are processed with larger patches.
* Extensive experimental results are provided.

**Weaknesses:**

* Lack of theoretical analysis
* Lack of computational analysis

**Questions:**

How is the adaptive patch sizes work with other transformer models, such as for example the Swin transformer.
Z. Liu et al. Swin Transformer: Hierarchical Vision Transformer using Shifted Windows, ICCV 2021

Can other methods be used of detecting complex regions for using smaller patches?

---

> ### Author Response · Authors · 2025-11-24
> **Response to Reviewer sF7Q**
>
> We are grateful for the positive review, and are glad the reviewer found our work to achieve the stated goal of increased efficiency while preserving performance.
>
> __Weakness: Lack of Computational Analysis__
>
> We appreciate this feedback! The main text of our submission included some runtime analysis in Table 6, which profiles the overhead of the APT method, and our main results include the throughput and GFLOPs of all models used. However, we have added more details in the main text about the runtime, with timing details on the sequence packing and patch embedding with ZeroConv in Figure 6, and how we compute the entropy on the CPU during dataloading.
>
> __Question: Can it be applied to Swin Transformer?__
>
> This is a great question! Our results on object detection and segmentation are explicitly applied to Swin-Transformer architectures. We mentioned this in Lines 244-251 on Page 5, but have updated the text to make this clearer, with our changes marked in blue. To handle window attention, we first restrict window sizes to be multiples of the largest patch size. Then, we apply the APT patch assignment, resulting in different numbers of tokens within each window. Like in the basic case, this can be natively handled with sequence packing and FlashAttention2, resulting in minimal overhead.
>
> __Question: Can other methods be used to detect patch sizes?__
>
> This is a good point, and other methods besides entropy can be used. In our supplement, we tested the effect of using two other scorers: the Laplacian filter and an “upsampling-based” measure. We show that these methods perform quite similarly, but that entropy edges the others out at higher thresholds. We include visual comparisons in Figure 9 of the Appendix as well as a Pareto-curve measurement in Figure 7.
>
>
>
> Please let us know if we have addressed your concerns, and feel free to ask more questions as needed. Thank you!

---

### Author Response · Authors · 2025-11-24
**Reviewer Response: General**

Dear Reviewers: Thank you all for your comprehensive and clear feedback. We are glad that our paper was found to achieve strong results, with large speed-ups on downstream tasks, as well as containing clear and comprehensive experiments. We have updated the submission to include updates requested by individual reviewers, with edits marked in blue. We outline some common concerns and updates below.

__Project Page and Release.__

We have added an anonymous project page with an interactive visualization tool at https://clamsoup97.github.io/anonymous-projects/apt/. Our anonymized code is also available at https://github.com/clamsoup97/apt. This contains several visualizations of our results as well as a failure case on object detection.

__Common Concern: Alternative Scorers.__

We have included an ablation that measures the effects of alternative scoring mechanisms besides entropy in the Appendix, Lines 1023-1077, and we encourage reviewers to read our updated text, quantitative and qualitative analysis.

Thank you all for your comments, and please let us know if we can address any further concerns. We look forward to more productive discussion!

---

### Meta-Review · Area_Chair_izJZ · 2026-01-23

**Summary:**

Before the rebuttal, the paper had diverging scores, with highly positive and negative reviews. The major weaknesses mentioned by the reviewers were a lack of comparison to existing methods like MG-VIT, PPT (sSUC), lack of exploration of decision methods other than entropy (sF7q, u6bT), dependency on hyperparameters (sSUC, W17G), and a lack of Pareto optimal performance in some cases (u6bT).

**Reviewer Concerns:**

Most of the concerns have been addressed by the authors.

* **lack of comparison to existing methods like MG-VIT, PPT (sSUC):** This has been partially addressed by the authors' claim that related works require large re-training; however, it would still be useful to see inference time experiments on speedup, since the model training is only required once. Reviewer sSUC was satisfied with the author's response.

* ** lack of exploration of decision methods other than entropy (sF7q, u6bT):** sF7q and u6bT’s insight and the results (figure 9) that the patch resizing decision can be made fairly well, multiple good decision functions are fairly important and should be discussed in the main paper rather than the supplementary. But the concerns have been addressed.

* **dependency on hyperparameters (sSUC, W17G):** The reviewers are satisfied with the effect of hyperparameter visualization and the stability across a window. I would also recommend the author to show that this window of stability holds for other domains such as remote sensing data and medical images (that can be fairly low entropy or high entropy), compared to natural images.

* ** Lack of Pareto optimal performance in some cases (u6bT):** The authors agree with the reviewers and could make their claims a bit softer in the vision-language and detection case.

**Reviewer Scores:**

Both of the negative-rating reviewers interacted with the authors before the discussion period ended.

sF7q: the rating stays at already positive 8.
sSUC: already mentioned they are increasing the rating. Therefore 6.
u6bT: Some issues, such as a lack of a complete Pareto curve, remain outstanding, so the rating will likely remain at 6.
W17G: already addressed authors and changes ratings to 6.

---

### Decision · Program_Chairs · 2026-01-26

Accept (Poster)